# Approximating Shapley Explanations in Reinforcement Learning

**Daniel Beechey**
University of Bath
United Kingdom
djeb20@bath.ac.uk

**Özgür Şimşek**
University of Bath
United Kingdom
o.simsek@bath.ac.uk

## Abstract

Reinforcement learning has achieved remarkable success in complex decision-making environments, yet its lack of transparency limits its deployment in practice, especially in safety-critical settings. Shapley values from cooperative game theory provide a principled framework for explaining reinforcement learning; however, the computational cost of Shapley explanations is an obstacle for their use. We introduce FastSVERL, a scalable method for explaining reinforcement learning by approximating Shapley values. FastSVERL is designed to handle the unique challenges of reinforcement learning, including temporal dependencies across multi-step trajectories, learning from off-policy data, and adapting to evolving agent behaviours in real time. FastSVERL introduces a practical, scalable approach for principled and rigourous interpretability in reinforcement learning.

## 1   Introduction

Reinforcement learning has achieved remarkable success in complex decision-making environments [3, 25], but the lack of transparency in an agent's decisions limits its deployment in practice, especially in safety-critical settings. While various interpretability methods have been proposed [9, 19], they often lack theoretical guarantees. A principled and promising approach [1, 2] is the use of Shapley values [26] to attribute the influence of features—numerical values that describe what an agent observes in its environment—on the agent's *behaviour*, *outcomes*, and *predictions*. Grounding these attributions in Shapley values provides a principled method for fair, transparent, and consistent credit assignment. One significant difficulty is that the computational cost of the Shapley approach scales exponentially with the number of features, making its application impractical for most real-world problems. There is therefore a strong need to develop scalable approximation methods that enable practical Shapley-based interpretability in reinforcement learning.

In supervised learning, scalable approximations of Shapley values are achieved through two primary methods: sampling over feature subsets [29, 27, 28, 15] and parametric models that amortise computation across inputs [8, 12, 7]. These methods are designed primarily for single-step predictions, whereas reinforcement learning introduces temporal dependencies across sequences of decisions. For example, explaining an agent's *outcomes* requires attributing influence over its expected return, which accumulates across possibly an infinite number of decisions. Furthermore, (1) as an agent's behaviour evolves, the explanations must evolve with it, and (2) when environment interaction is limited, explanations must be approximated from off-policy data—gathered by agents acting differently from the agent being explained. Addressing these challenges requires approximation methods tailored to the multi-stage, interactive, and evolving nature of reinforcement learning. We develop such an approach here, which we call *FastSVERL*.[1]

---

[1]FastSVERL code is available at: https://github.com/djeb20/fastsverl.

We introduce a scalable parametric learning method for approximating Shapley values in reinforcement learning. The proposed approach is designed to handle the temporal dependencies inherent to reinforcement learning, amortising Shapley value estimation across multi-step trajectories to explain agent behaviour, outcomes, and predictions efficiently. We address the practical constraints introduced in real-world reinforcement learning by considering how to explain policies from off-policy data whilst adapting them to evolving agent behaviours. The comprehensive, model-based framework we introduce provides a solid foundation that the research community can build upon. We present one such promising extension in Section 5, which preliminary findings suggest halves computational costs, improves estimation accuracy, and extends naturally to supervised learning.

These contributions position FastSVERL as a principled, rigourous, and scalable solution for interpretability in the practice of reinforcement learning.

## 2  Background

**Reinforcement learning** [30] models an agent interacting with its environment to achieve desired outcomes. It is commonly formalised as a Markov Decision Process (MDP), defined by the 5-tuple $(\mathcal{S}, \mathcal{A}, p, r, \gamma)$. The environment is initialised in a state $S_0 \in \mathcal{S}$, following initial state distribution $d(s) := \Pr(S_0 = s)$. At each decision stage $t \geq 0$, the agent observes the environment's current state $S_t \in \mathcal{S}$, which we assume can be decomposed into $n$ features $(S_t^1, \ldots, S_t^n)$ indexed by $\mathcal{F} = \{1, \ldots, n\}$; the agent selects an action $A_t \in \mathcal{A}$; the environment then transitions to state $S_{t+1} \in \mathcal{S}$ according to the transition kernel $p(s' \mid s, a) := \Pr(S_{t+1} = s' \mid S_t = s, A_t = a)$, and emits a scalar reward $R_{t+1} \in \mathbb{R}$ with expected value $r(s, a) = \mathbb{E}[R_{t+1} \mid S_t = s, A_t = a]$. A *policy* $\pi : \mathcal{S} \to \Delta(\mathcal{A})$ maps each state to a distribution over actions. The agent's objective is to learn a policy that maximises the discounted sum of future rewards, known as *expected return*, which is captured by the *state value function*:

$$v^\pi(s) := \mathbb{E}_\pi [G_t] = \mathbb{E}_\pi \left[ \sum_{k=0}^{\infty} \gamma^k R_{t+k+1} \mid S_t = s \right], \tag{1}$$

where $\gamma \in [0, 1]$ is a discount factor that weights future rewards. An *optimal policy* is a policy that maximises $v^\pi(s)$ for all $s \in \mathcal{S}$. In practice, reinforcement learning agents can use algorithms such as Deep Q-Networks (DQN) [17] and Proximal Policy Optimisation (PPO) [24] to learn near-optimal policies and value functions using neural networks. However, such methods generally do not explain the process behind an agent's behaviour, even though such explanations are often needed in practice.

**Explaining supervised learning with Shapley values.** One approach to interpreting supervised learning models is to attribute their predictions to the influence of individual input features [28, 15, 6]. Consider a model $f : \mathcal{X} \to \mathbb{R}$ and an input $x = (x^1, x^2, \ldots, x^n) \in \mathcal{X}$ defined over a set of $n$ features $\mathcal{F} = \{1, \ldots, n\}$. To quantify how features contribute to the prediction $f(x)$, we can consider how the model's output changes when some features are unknown. A *characteristic function* $f_x(\mathcal{C})$ [28] represents the model's expected prediction when only the features in subset $\mathcal{C} \subseteq \mathcal{F}$ are known:

$$f_x(\mathcal{C}) = \mathbb{E} \left[ f(X) \mid X^{\mathcal{C}} = x^{\mathcal{C}} \right], \tag{2}$$

where $x^{\mathcal{C}}$ denotes a subset of values $\{x^i : i \in \mathcal{C}\}$. In particular, the difference $f_x(\mathcal{F}) - f_x(\emptyset)$ quantifies the total change in prediction when all features are known versus when none are known. A principled way to distribute this quantity among the features is via Shapley values [26], which assign credit to each feature based on its mean marginal contribution across all possible subsets of features:

$$\phi^i(f_x) = \sum_{\mathcal{C} \subseteq \mathcal{F} \setminus \{i\}} \frac{|\mathcal{C}|! \cdot (|\mathcal{F}| - |\mathcal{C}| - 1)!}{|\mathcal{F}|!} [f_x(\mathcal{C} \cup \{i\}) - f_x(\mathcal{C})]. \tag{3}$$

Shapley values give the unique solution that satisfies four axioms formalising the notion of fairly attributing a given prediction among features. Whilst they have strong theoretical guarantees, the cost of computing Shapley values grows exponentially with the number of features. Each characteristic value is an expectation over the input space $\mathcal{X}$, with complexity $\mathcal{O}(|\mathcal{X}|)$. Shapley values sum over $2^n$ such values, one for each subset of features, giving a total cost of $\mathcal{O}(2^n \cdot |\mathcal{X}|)$ per input. As a result, the characteristic values in Equation 2 and the Shapley value sum in Equation 3 must be approximated in high-dimensional domains.

**Approximating characteristic values.** One common approach to approximating the characteristic function $f_x(\mathcal{C})$ is to learn a parametric model $\hat{f}(x \mid \mathcal{C}; \beta)$ that maps an input $x$, with features not in $\mathcal{C}$

replaced by a value outside the support of $\mathcal{X}$, to an approximate characteristic value [8]. The model is trained to minimise the expected squared error:

$$\mathcal{L}(\beta) = \mathop{\mathbb{E}}_{p(x)} \mathop{\mathbb{E}}_{p(\mathcal{C})} \left| f(x) - \hat{f}(x \mid \mathcal{C}; \beta) \right|^2, \tag{4}$$

where $p(x)$ is the data distribution and $p(\mathcal{C})$ is any distribution defined over all feature subsets. This loss cannot reach zero: for a sampled $\mathcal{C}$, different inputs $x$ sharing masked representations $x^{\mathcal{C}}$ may correspond to different $f(x)$ values. The model cannot recover every target with only the features in $\mathcal{C}$, and instead learns to predict their mean—namely, the characteristic value $f_x(\mathcal{C})$.

**Approximating the Shapley value summation.** We now describe how to approximate the Shapley summation in Equation 3, assuming access to a (possibly approximate) characteristic function $f_x(\mathcal{C})$. One approach is to learn a parametric model $\hat{\phi}(x; \theta) : \mathcal{X} \to \mathbb{R}^n$ that predicts the Shapley values for all features of an input $x$. This is the approach taken by FastSHAP [12], which trains a model $\hat{\phi}(x; \theta)$ using an equivalent characterisation of the Shapley values for a fixed input $x$ as the solution to a constrained least squares problem:

$$\{\phi^i(f_x)\}_{i \in \mathcal{F}} = \mathop{\arg\min}_{\{\phi^i\}_{i \in \mathcal{F}} \in \mathbb{R}^n} \mathop{\mathbb{E}}_{\mathcal{C} \sim p(\mathcal{C})} \left| f_x(\mathcal{C}) - f_x(\emptyset) - \sum_{i \in \mathcal{C}} \phi^i \right|^2 \text{ s.t. } \underbrace{\sum_{i=1}^{n} \phi^i = f_x(\mathcal{F}) - f_x(\emptyset)}_{\text{Efficiency constraint}}. \tag{5}$$

Here, the distribution $p(\mathcal{C})$ samples subsets in proportion to the combinatorial weights used in the Shapley value formula:

$$p(\mathcal{C}) \propto \frac{n - 1}{\binom{n}{|\mathcal{C}|} \cdot |\mathcal{C}| \cdot (n - |\mathcal{C}|)}, \tag{6}$$

for $\mathcal{C} \subset \mathcal{F}$ where $0 < |\mathcal{C}| < n$. The *efficiency constraint* reflects the requirement that the total contribution across features must equal the change in prediction from observing all features versus none. FastSHAP implements this characterisation by training the model $\hat{\phi}(x; \theta)$ to minimise the expected loss of the least squares objective over the data distribution $p(x)$:

$$\mathcal{L}(\theta) = \mathop{\mathbb{E}}_{p(x)} \mathop{\mathbb{E}}_{p(\mathcal{C})} \left| f_x(\mathcal{C}) - f_x(\emptyset) - \sum_{i \in \mathcal{C}} \hat{\phi}^i(x; \theta) \right|^2.$$

Since the model is unconstrained, a correction term is added post hoc to enforce the efficiency constraint in Equation 5:

$$\phi^i(f_x) \approx \hat{\phi}^i(x; \theta) + \frac{1}{n} \left( f_x(\mathcal{F}) - f_x(\emptyset) - \sum_{j \in \mathcal{F}} \hat{\phi}^j(x; \theta) \right).$$

**Explaining reinforcement learning with Shapley values.** Unlike supervised learning, which typically focuses on single predictions, reinforcement learning concerns how an agent sequentially interacts with its environment to achieve desired outcomes. To understand these long-term interactions, it is helpful to distinguish between three explanatory targets [2]: *behaviour* (how an agent acts), *outcome* (the consequences of those actions), and *prediction* (estimates of those outcomes). *SVERL* (Shapley Values for Explaining Reinforcement Learning) [1, 2] formalises these elements by attributing them to state features. While outcomes can refer to many consequences of an agent's behaviour, SVERL focuses on expected return, a standard formalisation of outcome in reinforcement learning. For each element, SVERL defines a characteristic function over subsets of features $\mathcal{C} \subseteq \mathcal{F}$, which measures how the agent's action, expected return, or prediction of expected return changes when only the features in $\mathcal{C}$ are observed. By evaluating how these measurements change across subsets of features, SVERL computes Shapley values to attribute each explanatory element to individual features. We now describe each characteristic function and explanation, beginning with behaviour.

SVERL explains behaviour by measuring how each feature influences the agent's action choice. The behaviour characteristic function $\tilde{\pi}_s^a(\mathcal{C})$ is defined as the expected probability of selecting action $a$ in state $s$ when only the features in $\mathcal{C}$ are known:

$$\tilde{\pi}_s^a(\mathcal{C}) := \mathbb{E}\left[ \pi(S, a) \mid S^{\mathcal{C}} = s^{\mathcal{C}} \right] = \sum_{s \in \mathcal{S}^+} p^\pi(s \mid s^{\mathcal{C}}) \, \pi(s, a), \tag{7}$$

where $\mathcal{S}^+$ is the set of non-terminal states. The distribution $p^\pi(S \mid S^{\mathcal{C}} = s^{\mathcal{C}})$ is the conditional steady-state distribution: the probability that an agent following policy $\pi$ is in state $s$, given that it

observes $s^{\mathcal{C}}$. Shapley values (Equation 3) are computed over the behaviour characteristic function to attribute the change in action probability when all features are known versus none, $\tilde{\pi}_s^a(\mathcal{F}) - \tilde{\pi}_s^a(\emptyset)$, to the individual features of state $s$.

SVERL explains outcomes by measuring how each feature contributes to the agent's expected return. The outcome characteristic function $\tilde{v}_s^\pi(\mathcal{C})$ is defined as the expected return received from state $s$ when the agent's policy has access only to the features $s^{\mathcal{C}}$:

$$\tilde{v}_s^\pi(\mathcal{C}) \coloneqq \mathbb{E}_\mu\left[G_t \mid S_t = s\right], \tag{8}$$

where $\mu$ is a modified policy that selects actions using the behaviour characteristic function $\tilde{\pi}_s^a(\mathcal{C})$ when features are unknown in state $s$, and follows the original policy $\pi$ elsewhere. Shapley values computed over the outcome characteristic function attribute the change in expected return across all features, $\tilde{v}_s^\pi(\mathcal{F}) - \tilde{v}_s^\pi(\emptyset)$, to the individual features of state $s$.

SVERL explains prediction by measuring how each feature contributes to the agent's, or an observer's, prediction of the agent's expected return $\hat{v}^\pi(s)$, which estimates the true expected return $v^\pi(s)$. The prediction characteristic function $\hat{v}_s^\pi(\mathcal{C})$ function is defined as the predicted expected return from state $s$ when only the features in $\mathcal{C}$ are known:

$$\hat{v}_s^\pi(\mathcal{C}) \coloneqq \mathbb{E}\left[\hat{v}^\pi(S) \mid S^{\mathcal{C}} = s^{\mathcal{C}}\right] = \sum_{s \in \mathcal{S}^+} p^\pi(s \mid s^{\mathcal{C}})\,\hat{v}^\pi(s). \tag{9}$$

Shapley values computed over this function attribute the change in predicted expected return across all features, $\hat{v}_s^\pi(\mathcal{F}) - \hat{v}_s^\pi(\emptyset)$, to the individual features of state $s$.

## 3 Approximating Shapley values in reinforcement learning

SVERL provides a rigorous and comprehensive framework for explaining reinforcement learning agents, but exact computation is impractical for most real-world problems. Each characteristic value is an expectation over the state space $\mathcal{S}$, and Shapley values sum these values over all possible combinations of features $\mathcal{F}$, resulting in a total cost of $\mathcal{O}(2^{|\mathcal{F}|} \cdot |\mathcal{S}|)$ per explanation. In high-dimensional settings, exact computation is infeasible, necessitating scalable approximation methods for both (1) the Shapley value sum and (2) the characteristic functions that underpin each explanation.

### 3.1 Approximating the Shapley value summation in reinforcement learning

We begin by considering how to approximate the Shapley value sum (Equation 3). Given a characteristic function for each type of explanation, the computation proceeds identically, allowing a single approximation method to be used across all three explanation types.

In supervised learning, a common approach to estimate the Shapley value sum is via Monte Carlo sampling, averaging marginal contributions over randomly selected subsets $\mathcal{C} \subseteq \mathcal{F}$ [28, 15]. However, such sampling estimates would not generalise across states and would need to be recomputed from scratch whenever the agent's policy changes. As a result, this approach would be inefficient for repeated explanations of evolving policies. We therefore do not pursue it here. Instead, we propose learning a parametric model that estimates the Shapley value contributions for all states. Our approach amortises the approximation cost across states and enables continual refinement of explanations under changing policies. We illustrate the method using behaviour explanations; we provide analogous loss functions for outcome and prediction explanations in Appendix A.

Specifically, we learn a parametric model $\hat{\phi}(s, a; \theta) : \mathcal{S} \times \mathcal{A} \to \mathbb{R}^{|\mathcal{F}|}$, with parameters $\theta$, to estimate the Shapley contributions of all features of state $s$ to the probability of selecting action $a$. We refer to models that predict Shapley values as *Shapley models*. Following FastSHAP [12], we adopt the characterisation of Shapley values in Equation 5 as the solution to a weighted least-squares problem, and train the Shapley model by minimising this loss:

$$\mathcal{L}(\theta) = \mathop{\mathbb{E}}_{p^\pi(s)} \mathop{\mathbb{E}}_{\text{Unif}(a)} \mathop{\mathbb{E}}_{p(\mathcal{C})} \left| \tilde{\pi}_s^a(\mathcal{C}) - \tilde{\pi}_s^a(\emptyset) - \sum_{i \in \mathcal{C}} \hat{\phi}^i(s, a; \theta) \right|^2. \tag{10}$$

Here, $p^\pi(s)$ denotes the steady-state distribution of the policy that the characteristic function $\tilde{\pi}_s^a(\mathcal{C})$ is defined over. The distribution $\text{Unif}(a)$ represents a uniform distribution over actions, and subsets $\mathcal{C}$ are drawn from the distribution in Equation 6. Because $p^\pi(s)$ is not directly accessible, we approximate the expectation by sampling from states encountered while following $\pi$. Since the model

is unconstrained, we add a post hoc correction to enforce the efficiency constraint in Equation 5:

$$\phi^i(\tilde{\pi}_s^a) \approx \hat{\phi}^i(s, a; \theta) + \frac{1}{|\mathcal{F}|}\left(\pi(s, a) - \tilde{\pi}_s^a(\emptyset) - \sum_{j \in \mathcal{F}} \hat{\phi}^j(s, a; \theta)\right). \tag{11}$$

In Appendix A, we prove that with a sufficiently expressive model class, the corrected output of the global optimum $\hat{\phi}(s, a; \theta^*)$ recovers exact and unbiased Shapley values for all state-action pairs $(s, a)$ such that $p^\pi(s) > 0$ and $a \in \mathcal{A}$. By substituting the appropriate characteristic function into the loss in Equation 10, the same model architecture and training procedure can be used across all SVERL explanations of behaviour, outcome, and prediction.

## 3.2 Approximating characteristic functions in reinforcement learning

SVERL explanations rely on a characteristic function defined as an expectation over the state space, which is infeasible to compute exactly in high-dimensional domains. We begin by considering how to approximate these functions for explaining behaviour and prediction, which share the same structure, before turning to explaining outcomes, which requires a different approach.

The **behaviour characteristic function** $\tilde{\pi}_s^a(\mathcal{C})$ in Equation 7 is the expected probability of taking action $a$ in state $s$, given only the features in subset $\mathcal{C}$. One possible approach is to estimate it via Monte Carlo sampling, drawing samples from the conditional steady-state distribution $p^\pi(s \mid s^\mathcal{C})$ [8]. However, such estimates would not generalise across states or feature subsets, making them inefficient for repeated explanations. We instead, we propose training a parametric model $\hat{\pi}(s, a \mid \mathcal{C}; \beta)$, with parameters $\beta$, to estimate the characteristic function, replacing features of $s$ not in $\mathcal{C}$ with a value outside the support of $\mathcal{S}$, minimising the expected squared error:

$$\mathcal{L}(\beta) = \mathbb{E}_{p^\pi(s)} \mathbb{E}_{\text{Unif}(a)} \mathbb{E}_{p(\mathcal{C})} |\pi(s, a) - \hat{\pi}(s, a \mid \mathcal{C}; \beta)|^2. \tag{12}$$

This approach amortises the approximation cost across multiple states and feature subsets. The model cannot recover the exact target $\pi(s, a)$ from partial input because different states can share the same values on a subset $\mathcal{C}$. It instead learns to predict their mean: the characteristic function $\pi_s^a(\mathcal{C})$. In Appendix B, we prove that with a sufficiently expressive model class, the global optimum $\hat{\pi}(s, a \mid \mathcal{C}; \beta^*)$ recovers the exact and unbiased characteristic values for all triples $(s, a, \mathcal{C})$ such that $p^\pi(s) > 0$, $a \in \mathcal{A}$, and $p(\mathcal{C}) > 0$.

The **prediction characteristic function** $\hat{v}_s^\pi(\mathcal{C})$ shares the same conditional expectation structure but uses $\hat{v}^\pi(s)$ as the prediction target. We propose approximating it using the same parametric modelling approach, minimising the expected squared error:

$$\mathcal{L}(\beta) = \mathbb{E}_{p^\pi(s)} \mathbb{E}_{p(\mathcal{C})} |\hat{v}^\pi(s) - \hat{v}(s \mid \mathcal{C}; \beta)|^2. \tag{13}$$

For the **outcome characteristic function** $\tilde{v}_s^\pi(\mathcal{C})$, defined as the expected return from a state $s_e$ when the agent follows a modified policy $\mu$ that selects actions using the behaviour characteristic $\tilde{\pi}_s^a(\mathcal{C})$ at $s_e$ and the original policy $\pi$ at all other states, we can use the parametric approximation of $\tilde{\pi}_s^a(\mathcal{C})$ from earlier, which reduces the challenge to estimating expected returns under the policy $\mu$ across many $(s_e, \mathcal{C})$ pairs.

The outcome characteristic $\tilde{v}_s^\pi(\mathcal{C})$ highlights the unique challenges posed by the sequential dynamics of reinforcement learning. For a fixed state $s_e$ and feature subset $\mathcal{C}$, the characteristic can be estimated using standard reinforcement learning techniques. However, estimating this quantity for all $(s_e, \mathcal{C})$ pairs amounts to solving $2^{|\mathcal{F}|} \times |\mathcal{S}|$ separate optimisation problems. To address this, we define a single conditioned policy $\hat{\pi}(a \mid s; s_e, \mathcal{C})$ that behaves according to $\tilde{\pi}_s^a(\mathcal{C})$ when the agent's current state $s$ matches the state-to-be-explained $s_e$, and otherwise follows the original policy $\pi$. This conditioned policy enables us to parametrise the agent's behaviour across all states $s$ and actions $a$ for any $(s_e, \mathcal{C})$ pair. We then introduce a parametric value function $V(s \mid s_e, \mathcal{C}; \beta)$ to estimate expected returns under the conditioned policy, amortising estimation of $\tilde{v}_s^\pi(\mathcal{C})$ across all $(s_e, \mathcal{C})$ pairs.

To estimate $V(s \mid s_e, \mathcal{C}; \beta)$, we consider the strengths and limitations of two general strategies from reinforcement learning. *Off-policy methods* can reuse data from earlier policies but may include few or no transitions under the policy $\hat{\pi}(a \mid s; s_e, \mathcal{C})$; *on-policy methods* avoid this difficulty by collecting new data from $\hat{\pi}$ at the cost of additional interaction with the environment. Both strategies have advantages; we present one representative approach for each. For clarity, we use simple DQN-style losses [17], but more advanced value-based methods can also be applied.

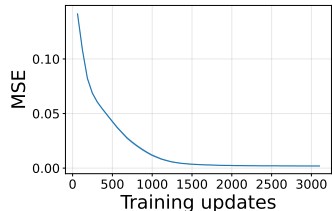
(a) Behaviour characteristic

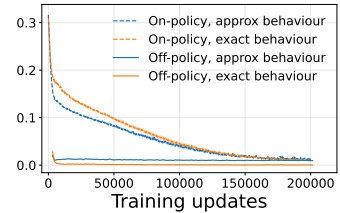
(b) Outcome characteristic

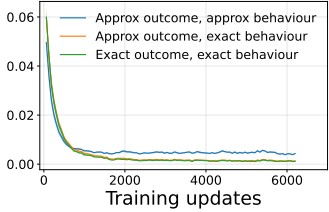
(c) Outcome Shapley values

Figure 1: How approximation accuracy improves with training updates in `Mastermind-222`. Shaded regions, which are negligible, indicate standard error over 20 runs. As we progress from plot (a) to (b) to (c), downstream models use exact or approximate upstream models from earlier plots.

For on-policy learning, we minimise the difference between the current state-value estimate $V(s \mid s_e, \mathcal{C}; \beta)$ and a target computed from the bootstrapped return:

$$\mathcal{L}(\beta) = \mathop{\mathbb{E}}_{(s,r,s',s_e,\mathcal{C}) \sim \mathcal{B}} \left| r + \gamma V'(s' \mid s_e, \mathcal{C}; \beta^-) - V(s \mid s_e, \mathcal{C}; \beta) \right|^2, \tag{14}$$

where $\mathcal{B}$ contains transitions sampled from the conditioned policy, and $V'$ is a target network with periodically updated parameters. To populate $\mathcal{B}$, consider collecting a single transition at decision stage $t$: the agent is in state $s_t$, and separately, a state-to-be-explained $s_e$ and feature subset $\mathcal{C}$ are sampled (e.g. from a replay buffer and uniform distribution, respectively). Conditioning on the sampled $s_e$ and $\mathcal{C}$ collapses the single conditioned policy $\hat{\pi}(a \mid s; s_e, \mathcal{C})$ into a standard policy for this step. The agent takes an action $a_t$ based on this policy: if its current state $s_t$ happens to be the state being explained, $s_e$, it acts according to the behaviour characteristic model (Equation 12); otherwise, it acts according to its original policy $\pi$. This action produces a single transition $(s_t, a_t, r_{t+1}, s_{t+1})$, which is stored in buffer $\mathcal{B}$.

For off-policy learning, we instead learn a state-action value function $Q(s, a \mid s_e, \mathcal{C}; \beta)$ to bootstrap using actions sampled from the conditioned policy $\hat{\pi}(a \mid s; s_e, \mathcal{C})$ by optimising the following objective:

$$\mathcal{L}(\beta) = \mathop{\mathbb{E}}_{(s,a,r,s') \sim \mathcal{B}} \mathop{\mathbb{E}}_{a' \sim \hat{\pi}(\cdot \mid s')} \mathop{\mathbb{E}}_{p^\pi(s_e)} \mathop{\mathbb{E}}_{p(\mathcal{C})} \left| r + \gamma Q'(s', a' \mid s_e, \mathcal{C}; \beta^-) - Q(s, a \mid s_e, \mathcal{C}; \beta) \right|^2. \tag{15}$$

Here, the buffer $\mathcal{B}$ contains transitions sampled from some other policy. The corresponding outcome characteristic is then recovered by:

$$\tilde{v}_s^\pi(\mathcal{C}) \approx \sum_{a \in \mathcal{A}} \hat{\pi}(s, a \mid s_e, \mathcal{C}) \cdot Q(s, a \mid s_e, \mathcal{C}; \beta). \tag{16}$$

Together, these approximation techniques form the basis of a scalable framework for generating Shapley-based explanations in reinforcement learning. We call this framework *FastSVERL*.

### 3.3 Empirical illustration

We illustrate the use of FastSVERL in multiple domains, guided by three questions on accuracy, efficiency, and scalability: (1) How well can the proposed models learn to approximate characteristic functions and Shapley values? (2) How many training updates are required to reach a given level of approximation error? (3) How does the computational cost of the approximation scale with the number of states and features in an environment?

We start by focusing on outcome explanations, a natural choice because they depend on three components—the behaviour characteristic, the outcome characteristic, and Shapley values—that together span the key parametric models and loss functions introduced in Sections 3.1 and 3.2. We approximate all three components for a DQN agent in the `Mastermind-222` domain used by Beechey et al. [2]. In the main paper, we present experiments on a subset of explanation types and domains, focusing on `Mastermind-222`, with eight features and 53 states, due to its tractability for exact Shapley value computation. In the appendix, analogous results for all explanation types, additional domains, and complete domain descriptions are provided, including experiments in larger domains where exact computation remains feasible only for behaviour and prediction explanations.

Figure 1 shows the mean squared error (MSE) between predicted and exact values for (a) the behaviour characteristic, (b) outcome characteristic, and (c) outcome Shapley values, averaged over

all states and features, plotted against training updates. For the outcome characteristic, we include both on-policy and off-policy training regimes. For all downstream models, we present results using exact or approximate upstream components (e.g. the outcome characteristic trained using the exact or approximate behaviour characteristic), to reveal how errors propagate through model approximations.

All three models converge to low approximation error ($< 0.01$). As expected, downstream models that rely on approximated characteristics—rather than exact ones—converge to less accurate solutions, illustrating how errors in upstream components degrade downstream performance. For the outcome characteristic, off-policy training converges faster than on-policy training, which is expected given that it reuses the original agent's training experience rather than collecting new transitions from the conditioned policy. Whilst the improvement is substantial, off-policy learning is effective only when the stored experience sufficiently covers states relevant to the conditioned policy.

We now examine how FastSVERL's approximations scale with domain size in `Hypercube`, an $n$-dimensional gridworld where the cube's dimension $n$ and side length $l$ control the number of states ($l^n$) and features ($n$). Figure 2 shows how the training updates required to reach a target loss of $0.01$ for the behaviour characteristic, and corresponding Shapley values, scales with the number of states and features. To isolate how the Shapley model scales without errors propagating from the behaviour characteristic, we train it using exact characteristic values.

For a fixed number of features (i.e. cube dimension), the training cost increases roughly linearly on the log-log scale, indicating approximate polynomial growth as the number of states increases. In contrast, given a fixed number of states, increasing the number of features has little effect on training cost, suggesting that states—not features—are the dominant driver of computational cost. Finally, the behaviour characteristic requires more updates than the Shapley model, possibly because it must predict values for all feature subsets at each state ($|\mathcal{S}| \times 2^n$). In contrast, the Shapley model predicts values per feature and state ($|\mathcal{S}| \times n$).

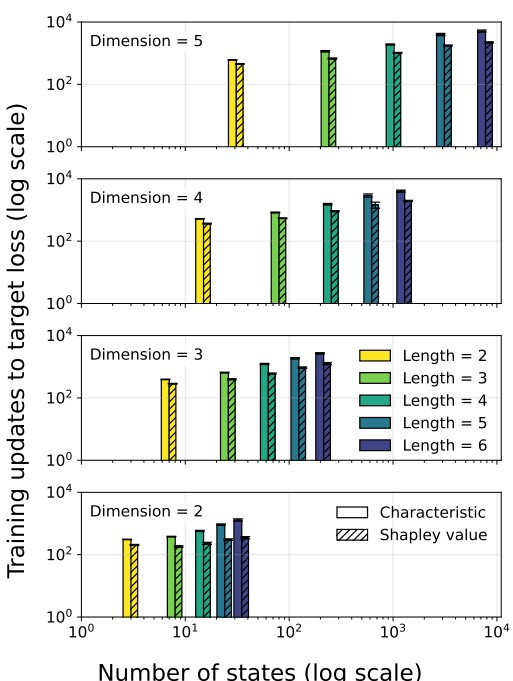

Figure 2: Training updates (mean $\pm$ standard error over 20 runs) needed to reach a fixed target loss (0.01) when approximating behaviour explanations in `Hypercube`. Each subplot fixes the number of features $n$ (i.e. dimensions). Bar colour indicates cube side length $l$.

Table 1: Convergence of behaviour models in large-scale `Mastermind` domains over 10 runs.

| Domain | Model | Updates to Converge (Mean $\pm$ Std. Err.) | Final Loss (Mean $\pm$ Std. Err.) |
|---|---|---|---|
| Mastermind-443 (24 features, $\geq 4.3 \times 10^7$ states) | Characteristic | $(1.10 \pm 0.11) \times 10^6$ | $(3.83 \pm 0.02) \times 10^{-3}$ |
| | Shapley | $(7.31 \pm 0.68) \times 10^5$ | $(1.30 \pm 0.04) \times 10^{-3}$ |
| Mastermind-453 (30 features, $\geq 3.5 \times 10^9$ states) | Characteristic | $(1.29 \pm 0.11) \times 10^6$ | $(3.60 \pm 0.02) \times 10^{-3}$ |
| | Shapley | $(7.84 \pm 0.49) \times 10^5$ | $(0.96 \pm 0.03) \times 10^{-3}$ |
| Mastermind-463 (36 features, $\geq 2.8 \times 10^{11}$ states) | Characteristic | $(1.18 \pm 0.12) \times 10^6$ | $(3.70 \pm 0.01) \times 10^{-3}$ |
| | Shapley | $(7.12 \pm 0.51) \times 10^5$ | $(1.88 \pm 0.04) \times 10^{-3}$ |

While *accuracy* validation requires small domains with computable ground truths, it is also important to investigate FastSVERL's approximations at scale. We now examine three key properties in substantially larger `Mastermind` domains: (1) if the models converge, (2) the stability of that convergence, and (3) if the computational cost remains manageable. Table 1 shows that both the characteristic and Shapley models converge and that this convergence is stable and reliable across runs. Most importantly, the number of training updates required remains consistent even as the number of states and features grows. Note that this training loss is not a measure of ground-truth

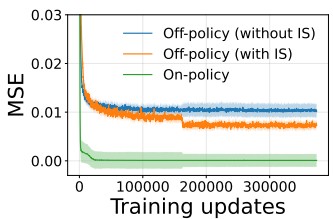
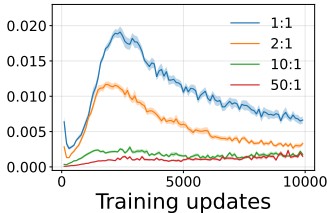
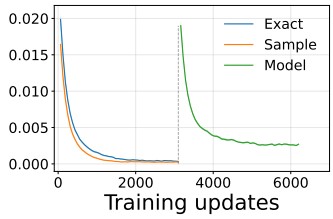

(a) Behaviour characteristic
(off-policy training)

(b) Prediction Shapley
(parallel training)

(c) Behaviour Shapley
(sampled characteristic)

Figure 3: Approximation accuracy over training updates in `Mastermind-222`. Each line shows the mean squared error (MSE) between predicted and exact values, averaged over all states and features. Shaded regions indicate standard error over 20 runs, corrected for variance in agent training [16].

accuracy. For the characteristic model, the loss is designed to converge to a non-zero value as it learns an expectation over unknown features. For the Shapley model, convergence to zero indicates it has learned to explain the *approximations* from the characteristic model, not necessarily the true values.

While ground-truth accuracy cannot be validated, we demonstrate example behavioural insights the framework can produce by visualising an explanation for a trained `Mastermind-463` agent in Table 2. In this version of the code-breaking game, an agent must guess a hidden 4-letter code, drawn from a 3-letter alphabet. Each guess receives clues for the number of correct letters in the correct position (Clue 2) and wrong position (Clue 1, full details in Appendix G). The table shows a board state where darker blue indicates a feature's positive contribution to the probability of the agent's next action (in green).

Taking care not to over-interpret [2], the explanation appears to align with a logical, high-level strategy for the game. First, the most influential features (blue cells) are from recent guesses (2-4), which are sufficient to deduce that the code is a permutation of (B, C, C, C), while the now-redundant first guess has a neutral influence. Secondly, the unused guess slots are correctly assigned a neutral influence. This reveals an important insight about the agent's behaviour: the optimal policy does not require these slots. Importantly, it also suggests the approximation satisfies the nullity axiom of Shapley values by assigning zero contribution to irrelevant features.

Table 2: Behaviour Shapley values in one sample state of Mastermind-463.

| Guess | Clue 1 | Pos 1 | Pos 2 | Pos 3 | Pos 4 | Clue 2 |
|-------|--------|-------|-------|-------|-------|--------|
| 6     |        |       |       |       |       |        |
| 5     |        | C     | C     | C     | B     |        |
| 4     | 2      | B     | C     | C     | C     | 2      |
| 3     | 2      | C     | B     | C     | C     | 2      |
| 2     | 0      | C     | C     | C     | C     | 3      |
| 1     | 0      | A     | A     | C     | A     | 1      |

## 4  Applying FastSVERL to common reinforcement learning settings

We now discuss approximating Shapley explanations under practical constraints.

**Learning to explain off-policy.** FastSVERL trains characteristic and Shapley models using samples from the steady-state distribution of the policy being explained. This distribution is approximated using states encountered by the policy. What if the agent cannot interact with the environment to collect this data, or if the interaction is costly? We then have a more constrained off-policy problem, distinct from the setting previously discussed for the performance characteristic, which addressed sample-efficient data reuse when new interactions are possible. Here, we propose learning passively by drawing states from the agent's history of interaction and applying importance sampling [18] to correct the distributional mismatch. For any loss of the form $\mathcal{L}(\beta) = \mathbb{E}_{s \sim p^\pi}[\ell(s; \beta)]$, we estimate:

$$\mathcal{L}(\beta) \approx \mathop{\mathbb{E}}_{(s_t, a_t) \sim \mathcal{B}} \left[ \frac{\pi(s_t, a_t)}{\pi_t(s_t, a_t)} \cdot \ell(s_t; \beta) \right], \tag{17}$$

where $\pi_t$ denotes the policy used at decision stage $t$ to generate sample $(s_t, a_t)$. The reweighting in Equation 17 estimates the loss that would have been observed had the data been collected under policy $\pi$. The usefulness of this estimate depends on factors such as whether the buffer contains states visited under $\pi$. We provide a full derivation and discussion of Equation 17 in Appendix D.

We illustrate the impact of importance sampling by training the behaviour characteristic model in `Mastermind-222` under three conditions: (1) using on-policy data from the final policy, (2) using off-policy data from the agent's training buffer without importance sampling (IS), and (3) using the same buffer with importance sampling. The results are shown in Figure 3a. Importance sampling lowers approximation error when using the agent's training buffer, as expected, but does not match the accuracy of the on-policy baseline.

**Continuously learning to explain.** FastSVERL explains an agent's behaviour, outcomes, and prediction under a given policy. Yet, in many settings, such as continual learning, we will want to explain an agent as its policy changes over time. Rather than retraining FastSVERL's models when the policy changes, we ask whether jointly updating them with the agent's policy can keep explanations aligned to the policy throughout learning. FastSVERL's approximations can be continuously updated because they use parametric models; in contrast, sampling-based methods would require recomputing explanations from scratch whenever the policy changes. To this end, we propose a training regime in which the interaction data used to update the agent's policy is also used to learn and continuously update the Shapley explanation models.

Once again, we find ourselves in an off-policy context because earlier behaviour policies will have typically generated the data. We therefore apply the importance sampling technique from Equation 17 to account for off-policy samples. When policy changes are small, as is typical when using algorithms like PPO [24], we expect the explanation models from the previous decision stage to remain closely aligned with the agent's newly updated policy. As a result, jointly updating the explanation models alongside the agent's policy may restore complete alignment.

We illustrate the proposed approach by jointly training a DQN agent, a characteristic model, and a Shapley model for prediction in `Mastermind-222`. To control how quickly the explanations adapt to the changing policy, we vary the number of gradient descent updates applied to the explanation models, per policy update, using the ratios 1:1, 2:1, 10:1, and 50:1. Figure 3b shows the approximation error in the Shapley model throughout training for the various update ratios tested. All configurations start with a low error for the agent's initial (random) policy because actions and value estimates are independent of features; the ground truth contributions already match the near-zero predictions of the randomly initialised Shapley model. As the agent's policy improves, the approximation error spikes for low update ratios (1:1 and 2:1). These spikes coincide with a sharp rise in expected return observed during training (see Appendix E), reflecting large policy updates and suggesting that explanation models aligned with the agent's previous policy become misaligned when the policy shifts significantly. However, increasing the update ratio mitigates these spikes: the 10:1 and 50:1 configurations maintain relatively low approximation error throughout. These results suggest that jointly training FastSVERL's models with the agent may allow explanations to remain aligned with a changing policy, provided the update rate is sufficient to track larger shifts.

## 5 Approximating Shapley values without characteristic models

We introduced FastSVERL, a model-based framework to approximate Shapley values in reinforcement learning, providing a solid foundation upon which the community can build. Here, we present one such promising extension. FastSVERL uses parametric approximations of characteristic functions to train Shapley models. Training these characteristic models is a major computational bottleneck, with the `Hypercube` experiment in Figure 2 suggesting they require as much computation as training the Shapley models themselves. This motivates the search for alternative approximation methods. Monte Carlo sampling might be an appealing alternative because it removes the need for parametric models entirely. However, this merely shifts the computational burden from model training to the high cost of repeated sampling. What if we could preserve the model-free benefits of sampling without incurring its computational cost? We propose integrating single-sample approximations of characteristic values (with low computational cost) directly into the Shapley model loss, amortising characteristic estimation within Shapley training itself. We illustrate this idea using behaviour explanations.

We replace the behaviour characteristic $\tilde{\pi}_s^a(\mathcal{C})$ in the Shapley model loss (Equation 10) with a single-sample approximation. At each loss evaluation, we sample a state $s' \sim p^\pi(\cdot \mid s^\mathcal{C})$ and use the action probability $\pi(s', a)$ in place of querying a characteristic model:

$$\mathcal{L}(\theta) = \mathop{\mathbb{E}}_{p^\pi(s)} \mathop{\mathbb{E}}_{\text{Unif}(a)} \mathop{\mathbb{E}}_{p(\mathcal{C})} \mathop{\mathbb{E}}_{s' \sim p^\pi(\cdot \mid s^\mathcal{C})} \left| \pi(s', a) - \pi_{s,a}(\emptyset) - \sum_{i \in \mathcal{C}} \hat{\phi}^i(s, a; \theta) \right|^2 . \tag{18}$$

In Appendix F, we prove that optimising this new loss recovers the exact and unbiased Shapley values, with the same per-update cost as the original loss in Equation 10. This approach presents a favourable trade-off: we accept higher variance in individual gradients to eliminate the cost of pre-training a characteristic model and prevent its approximation errors from propagating into the Shapley model.

We illustrate the proposed approach by training a behaviour Shapley model in `Mastermind-222` with three characteristic approximations: (1) single-samples, (2) model-based, and (3) exact values. Figure 3c shows the Shapley model's loss over cumulative training updates, reflecting characteristic and Shapley model training. The model-based curve starts later because its initial updates are dedicated to learning the characteristic model (indicated by the dashed line), and it converges to a higher loss, reflecting approximation errors in the characteristic. In contrast, sampling converges before the model-based setup begins its Shapley updates, matching the performance of using *exact characteristic values*. Interestingly, the model trained with exact values converges slightly slower. As observed in prior work [7], the noise from the single-sample approximations may act as a stochastic regulariser, encouraging more efficient learning by preventing the model from overfitting to a single batch. These results show how replacing characteristic models with sampling can improve Shapley model efficiency and accuracy: halving total training time and eliminating error propagation.

## 6 Related work

Shapley values have been applied to explain behaviour [21, 4, 10, 32, 13, 14, 20, 31] and prediction [34, 23, 35] in reinforcement learning. Beechey et al. [1, 2] studied the theoretical validity of Shapley values in reinforcement learning. Their analysis revealed outcomes as a missing explanatory element, unifying behaviour, outcomes, and prediction under a single theoretical framework, SVERL. We build on this earlier foundation to develop scalable approximation techniques, which make it possible to use SVERL in practical settings.

In supervised learning, Shapley values were initially approximated through sampling [29, 27, 28, 15]. More recent approaches improve efficiency by amortising computation across inputs with parametric models [8, 12] that incorporate noisy Shapley targets during training [7]. Our work incorporates the use of these amortised methods into reinforcement learning, addressing practical constraints such as off-policy data and continual learning. In addition, we introduce a single-sample noisy target that can eliminate the need for characteristic models, substantially improving training efficiency.

## 7 Discussion

We introduced FastSVERL, a scalable parametric framework for Shapley-based explanations of reinforcement learning. FastSVERL estimates Shapley values in a single forward pass, without relying on costly Monte Carlo samples. We addressed two practical challenges: (i) off-policy learning, to enable explanations with minimal environment interaction, and (ii) continual learning, to allow explanations of non-stationary policies in real time. We showed that replacing characteristic models with single-sampled approximations can substantially improve efficiency and explanation quality; this approach also directly extends to supervised learning. These contributions position FastSVERL as a scalable solution for interpretability in practical reinforcement learning problems.

The proposed methods are broadly applicable to episodic and continuous tasks, as well as partially observable settings. Although this work focuses on discrete action spaces, the underlying theoretical framework supports continuous actions, and FastSVERL's parametric approach readily adapts. Applying the framework to continuous state spaces, however, highlights a key challenge: approximating the steady-state distribution. In high-dimensional settings, experience buffers provide only a sparse approximation of the true distribution. One promising direction is to learn a parametric model of this distribution [8], which could generalise across sparsely sampled regions.

Our work focused on the significant computational challenges of approximating Shapley values. Formal user studies and practical deployment in real-world systems are essential to evaluate how these explanations aid human understanding and decision-making. While our qualitative results suggest the framework can produce plausibly interpretable insights, user studies can rigorously evaluate robustness, identify use cases, and reveal the conditions under which the approximations become unreliable. This next step is critical for establishing best practices and strengthening FastSVERL's capacity for scalable, real-time interpretability in complex reinforcement learning environments.

## Acknowledgments and Disclosure of Funding

This work was supported by the Engineering and Physical Sciences Research Council (EPSRC) [grant number EP/X025470/1] and the UKRI Centre for Doctoral Training in Accountable, Responsible and Transparent AI (ART-AI) [EP/S023437/1]. We thank our reviewers for a constructive process.

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

# A Shapley models

This section presents the learning objectives used to train the Shapley models for all three explanation types: behaviour, outcomes, and prediction. For each case, we adopt the characterisation of Shapley values as the solution to a weighted least squares problem (see Equation 5) and construct a corresponding loss function. A general convergence result applicable to all three is provided at the end of the section.

**Behaviour.** We train a parametric model $\hat{\phi}(s, a; \theta) : \mathcal{S} \times \mathcal{A} \to \mathbb{R}^{|\mathcal{F}|}$ to predict the Shapley contributions of each feature of state $s$ to the agent's probability of selecting action $a$. The model is trained to minimise the following loss:

$$\mathcal{L}(\theta) = \mathop{\mathbb{E}}_{p^\pi(s)} \mathop{\mathbb{E}}_{\mathrm{Unif}(a)} \mathop{\mathbb{E}}_{p(\mathcal{C})} \left| \tilde{\pi}_s^a(\mathcal{C}) - \tilde{\pi}_s^a(\emptyset) - \sum_{i \in \mathcal{C}} \hat{\phi}^i(s, a; \theta) \right|^2. \tag{19}$$

After training, the model output is corrected to enforce the efficiency constraint:

$$\phi^i(\tilde{\pi}_s^a) \approx \hat{\phi}^i(s, a; \theta) + \frac{1}{|\mathcal{F}|} \left( \pi(s, a) - \tilde{\pi}_s^a(\emptyset) - \sum_{j \in \mathcal{F}} \hat{\phi}^j(s, a; \theta) \right). \tag{20}$$

With a sufficiently expressive model class and exact optimisation, the corrected output of the global optimum $\hat{\phi}(s, a; \theta^*)$ recovers exact Shapley values for all state-action pairs $(s, a)$ such that $p^\pi(s) > 0$ and $a \in \mathcal{A}$.

**Outcome.** We train a parametric model $\hat{\phi}(s; \theta) : \mathcal{S} \to \mathbb{R}^{|\mathcal{F}|}$ to predict the Shapley contributions of each feature of state $s$ to the agent's expected return $v^\pi(s)$. The model is trained to minimise the following loss:

$$\mathcal{L}(\theta) = \mathop{\mathbb{E}}_{p^\pi(s)} \mathop{\mathbb{E}}_{p(\mathcal{C})} \left| \tilde{v}_s^\pi(\mathcal{C}) - \tilde{v}_s^\pi(\emptyset) - \sum_{i \in \mathcal{C}} \hat{\phi}^i(s; \theta) \right|^2. \tag{21}$$

After training, the model output is corrected to enforce the efficiency constraint:

$$\phi^i(\tilde{v}_s^\pi) \approx \hat{\phi}^i(s; \theta) + \frac{1}{|\mathcal{F}|} \left( v^\pi(s) - \tilde{v}_s^\pi(\emptyset) - \sum_{j \in \mathcal{F}} \hat{\phi}^j(s; \theta) \right). \tag{22}$$

With a sufficiently expressive model class, the corrected output of the global optimum $\hat{\phi}(s; \theta^*)$ recovers exact Shapley values for all states $s$ such that $p^\pi(s) > 0$.

**Prediction.** We train a parametric model $\hat{\phi}(s; \theta) : \mathcal{S} \to \mathbb{R}^{|\mathcal{F}|}$ to estimate the Shapley contributions of each feature of state $s$ to a prediction of the agent's expected return $\hat{v}^\pi(s)$. The model is trained to minimise the following loss:

$$\mathcal{L}(\theta) = \mathop{\mathbb{E}}_{p^\pi(s)} \mathop{\mathbb{E}}_{p(\mathcal{C})} \left| \hat{v}_s^\pi(\mathcal{C}) - \hat{v}_s^\pi(\emptyset) - \sum_{i \in \mathcal{C}} \hat{\phi}^i(s; \theta) \right|^2. \tag{23}$$

After training, the model output is corrected to enforce the efficiency constraint:

$$\phi^i(\hat{v}_s^\pi) \approx \hat{\phi}^i(s; \theta) + \frac{1}{|\mathcal{F}|} \left( \hat{v}^\pi(s) - \hat{v}_s^\pi(\emptyset) - \sum_{j \in \mathcal{F}} \hat{\phi}^j(s; \theta) \right). \tag{24}$$

With a sufficiently expressive model class, the corrected output of the global optimum $\hat{\phi}(s; \theta^*)$ recovers exact Shapley values for all states $s$ such that $p^\pi(s) > 0$.

**Convergence proof.** We now prove that the learning objective in Equation 19 recovers exact Shapley values at the global optimum. This result generalises to the prediction and outcome objectives because they share the same structure.

We make the following assumptions:

1. The model $\hat{\phi}(s, a; \theta)$ is selected from a function class expressive enough to represent the true Shapley value function $\phi(\tilde{\pi}_s^a)$ for all state-action pairs $(s, a)$ such that $p^\pi(s) > 0$.

2. The global minimum of the loss $\mathcal{L}(\theta)$ exists and is attained.

3. States $s$ are sampled from the steady-state distribution $p^\pi(s)$, and actions are sampled uniformly from $\mathcal{A}$.

Fix a state-action pair $(s, a)$ such that $p^\pi(s) > 0$. The learning objective for this pair reduces to the following expected loss:

$$\mathbb{E}_{p(\mathcal{C})}\left[\left(\tilde{\pi}_s^a(\mathcal{C}) - \tilde{\pi}_s^a(\emptyset) - \sum_{i \in \mathcal{C}} \phi^i\right)^2\right], \tag{25}$$

where $\phi \in \mathbb{R}^{|\mathcal{F}|}$ denotes the predicted attribution vector produced by the model for this fixed $(s, a)$.

If the following efficiency constraint is imposed:

$$\sum_{i \in \mathcal{F}} \phi^i = \tilde{\pi}_s^a(\mathcal{F}) - \tilde{\pi}_s^a(\emptyset), \tag{26}$$

then this becomes a constrained weighted least-squares problem with a unique global minimiser given by the Shapley values $\phi(\tilde{\pi}_s^a)$ [5].

This constraint can be satisfied by applying an additive correction to the model output:

$$\phi^i(\tilde{\pi}_s^a) := \hat{\phi}^i(s, a; \theta) + \frac{1}{|\mathcal{F}|}\left(\pi(s, a) - \tilde{\pi}_s^a(\emptyset) - \sum_{j \in \mathcal{F}} \hat{\phi}^j(s, a; \theta)\right), \tag{27}$$

which adjusts the model output by a constant shift that distributes the residual error uniformly across features [22]. This correction preserves the minimiser of the original unconstrained loss because the transformation is linear and orthogonal to the residual, and ensures the efficiency constraint is satisfied.

Since the model class contains the exact solution and the loss is minimised exactly, the globally optimal parameters $\theta^*$ yield a model $\hat{\phi}(s, a; \theta^*)$ that recovers the Shapley values for all $(s, a)$ such that $p^\pi(s) > 0$ and $a \in \mathcal{A}$. ∎

**Remark.** All of the Shapley models are trained to solve a weighted least-squares problem whose unique solution is the true Shapley value vector for the given characteristic function. Therefore, these estimators are asymptotically unbiased.

## B  Characteristic models

This section presents the learning objectives and convergence results for training the characteristic models for behaviour, prediction, and outcome explanations.

**Behaviour.** We train a parametric model $\hat{\pi}(s, a \mid \mathcal{C}; \beta)$ to approximate the characteristic function $\tilde{\pi}_s^a(\mathcal{C})$: the expected action-probability $\pi(s, a)$ when only the features in $\mathcal{C}$ are known. The model receives as input a state-action pair $(s, a)$, where features of $s$ not in $\mathcal{C}$ are replaced by a fixed masking value outside the support of $\mathcal{S}$. It is trained to minimise the following loss:

$$\mathcal{L}(\beta) = \mathbb{E}_{p^\pi(s)} \mathbb{E}_{\text{Unif}(a)} \mathbb{E}_{p(\mathcal{C})} |\pi(s, a) - \hat{\pi}(s, a \mid \mathcal{C}; \beta)|^2. \tag{28}$$

With a sufficiently expressive model class and exact optimisation, the global optimum $\hat{\pi}(s, a \mid \mathcal{C}; \beta^*)$ recovers the exact characteristic value for all triples $(s, a, \mathcal{C})$ such that $p^\pi(s) > 0$, $a \in \mathcal{A}$ and $p(\mathcal{C}) > 0$:

$$\hat{\pi}(s, a \mid \mathcal{C}; \beta^*) = \mathbb{E}\left[\pi(S, a) \mid S^\mathcal{C} = s^\mathcal{C}\right] = \tilde{\pi}_s^a(\mathcal{C}). \tag{29}$$

**Prediction.** We train a parametric model $\hat{v}(s \mid \mathcal{C}; \beta)$ to approximate the characteristic function $\hat{v}_s^\pi(\mathcal{C})$: the predicted expected return $\hat{v}^\pi(s)$ when only the features in $\mathcal{C}$ are known. The model receives as input a state $s$, where features not in $\mathcal{C}$ are replaced by a fixed masking value outside the support of $\mathcal{S}$. It is trained to minimise the following loss:

$$\mathcal{L}(\beta) = \mathbb{E}_{p^\pi(s)} \mathbb{E}_{p(\mathcal{C})} |\hat{v}(s) - \hat{v}(s \mid \mathcal{C}; \beta)|^2. \tag{30}$$

With a sufficiently expressive model class and exact optimisation, the global optimum $\hat{v}(s \mid \mathcal{C}; \beta^*)$ recovers the exact characteristic value for all pairs $(s, \mathcal{C})$ such that $p^\pi(s) > 0$ and $p(\mathcal{C}) > 0$:

$$\hat{v}(s \mid \mathcal{C}; \beta^*) = \mathbb{E}\left[\hat{v}(S) \mid S^\mathcal{C} = s^\mathcal{C}\right] = \hat{v}_s(\mathcal{C}). \tag{31}$$

**Behaviour convergence proof.** We now prove that the learning objective in Equation 28 recovers exact characteristic values at the global optimum. This result generalises to the prediction objective in Equation 30 because they share the same structure.

We make the following assumptions:

1. The model $\hat{\pi}(s, a \mid \mathcal{C}; \beta)$ is selected from a function class expressive enough to represent $\tilde{\pi}_s^a(\mathcal{C})$ for all masked inputs corresponding to $(s^\mathcal{C}, a)$ such that $p^\pi(s) > 0$ and $p(\mathcal{C}) > 0$.

2. The global minimum of the loss $\mathcal{L}(\beta)$ exists and is attained.

3. States $s$ are sampled from the steady-state distribution $p^\pi(s)$, actions from the uniform distribution over $\mathcal{A}$, and subsets from a fixed distribution over all subsets of $\mathcal{F}$.

We consider the contribution to the loss from a fixed subset $\mathcal{C}$ and action $a$. The full loss is an expectation over such terms. Under this conditioning, the loss reduces to:

$$\mathbb{E}_{p^\pi(s)}\left[\left|\pi(s, a) - \hat{\pi}(s, a \mid \mathcal{C}; \beta)\right|^2\right], \tag{32}$$

where $\mathcal{C}$ and $a$ are fixed. Because the model receives masked inputs in which features outside $\mathcal{C}$ are replaced by a fixed value, it produces the same output for all states $s$ that share the same values $s^\mathcal{C}$ on $\mathcal{C}$. As a result, the loss decomposes into disjoint terms over equivalence classes of $s^\mathcal{C}$:

$$\sum_{s^\mathcal{C}} p^\pi(s^\mathcal{C}) \mathbb{E}_{p^\pi(s|s^\mathcal{C})}\left[\left|\pi(s, a) - \hat{\pi}(s, a \mid \mathcal{C}; \beta)\right|^2\right], \tag{33}$$

where $p^\pi(s^\mathcal{C})$ denotes the marginal distribution over observed feature subsets under $p^\pi(s)$. Each term is a squared-error regression problem in which the model output is constant across the equivalence class. Therefore, the unique minimiser of each such term is the conditional expectation:

$$\hat{\pi}(s, a \mid \mathcal{C}; \beta^*) = \mathbb{E}\left[\pi(S, a) \mid S^\mathcal{C} = s^\mathcal{C}\right] = \tilde{\pi}_s^a(\mathcal{C}), \tag{34}$$

for all $s$ such that $p^\pi(s^\mathcal{C}) > 0$.

Since both the subset $\mathcal{C}$ and action $a$ were arbitrary, and the result holds for all equivalence classes $s^\mathcal{C}$ with $p^\pi(s^\mathcal{C}) > 0$, the global minimiser of the full loss recovers exact characteristic values for all $(s, a, \mathcal{C})$ such that $p^\pi(s) > 0$ and $p(\mathcal{C}) > 0$. ∎

**Remark.** The characteristic models for behaviour and prediction are trained to minimise an expected squared-error loss. The unique global minimiser of this objective is the true conditional expectation. An estimator that converges to this true mean is, by definition, asymptotically unbiased.

**Outcome characteristic convergence.** We do not provide a convergence proof for the outcome characteristic model. This component is trained as a value function using standard reinforcement learning techniques, including bootstrapping and temporal-difference updates. In our case, the implementation is based on the DQN algorithm. While convergence guarantees exist for value-based learning in the tabular setting [33], convergence results are generally not known for deep reinforcement learning methods with function approximation. This remains a long-standing open problem in the field. As such, the convergence behaviour of the outcome characteristic model cannot be formally established. Nonetheless, we empirically observe reliable learning across all domains this work considers.

## C   Empirical illustrations of FastSVERL's explanation models

In this section, we extend the empirical illustration of FastSVERL presented in Figures 1 and 2 by providing results for the complete set of explanation types—behaviour, outcomes, and prediction.

**Accuracy.** Figure 4 on the following page shows approximation accuracy as a function of batch updates of gradient descent (training updates) for characteristic and Shapley models applied to a DQN agent across three domains: `Mastermind-222` (8 features, 53 states), `Mastermind-333` (15 features,

over 100,000 states), and `Gridworld` (2 features, 7 states). Complete domain descriptions are provided in Appendix G. Notably, `Mastermind-333` represents the practical limit of exact Shapley value computation for behaviour and prediction explanations, highlighting FastSVERL's ability to approximate these explanations in substantially larger state spaces.

Each experimental run is seeded for reproducibility, with variability primarily stemming from sources typical to PyTorch-based training, including weight initialisation and batch sampling. Characteristic models are trained using a single agent instance across all runs. Likewise, Shapley models are always trained with the same agent and characteristic models for each run, avoiding variability introduced by retraining upstream components.

Consistent with the findings in Section 3.3, all models converge to low approximation error and exhibit clear error propagation from characteristic models to Shapley values.

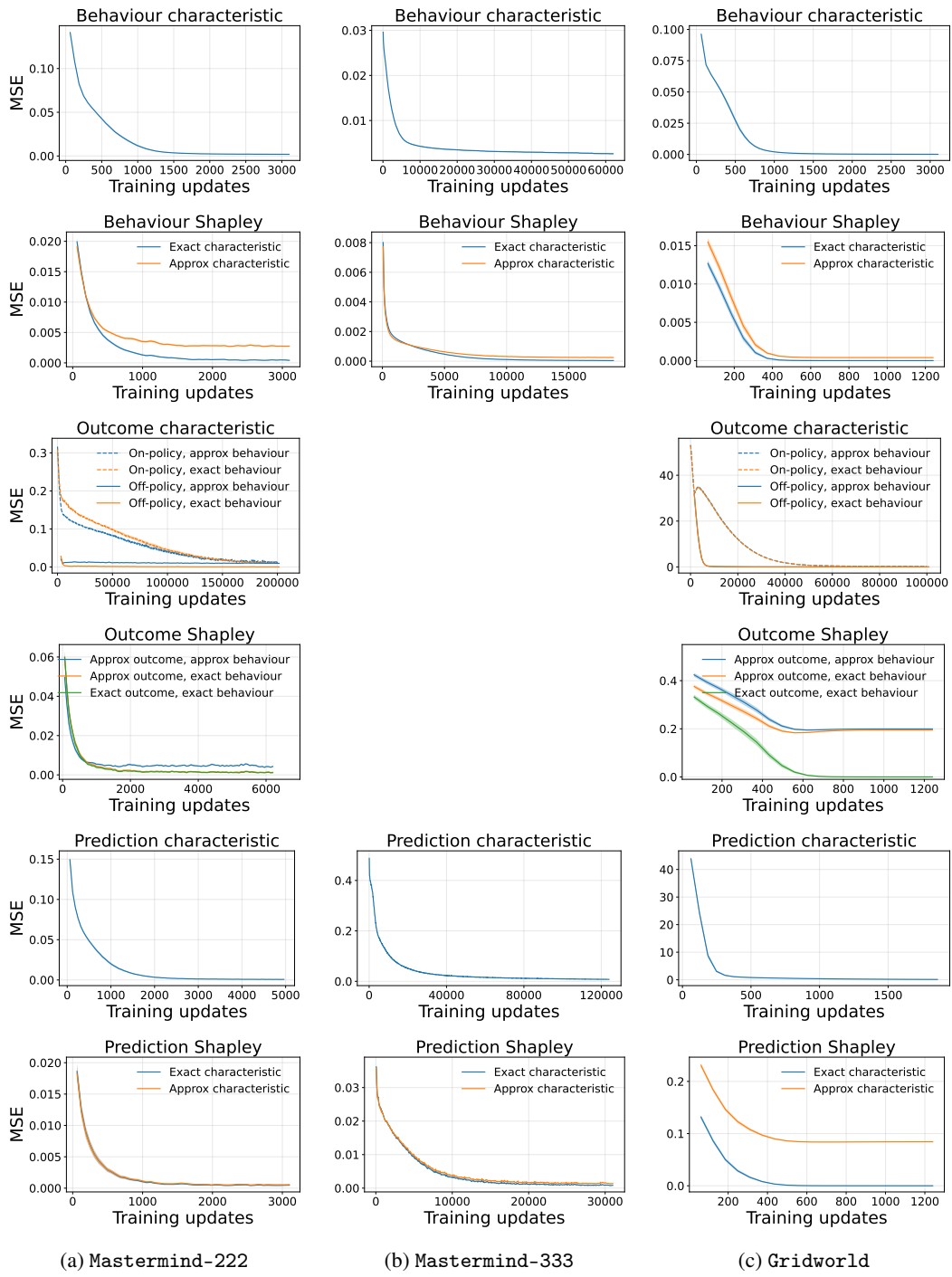

(a) `Mastermind-222`        (b) `Mastermind-333`        (c) `Gridworld`

Figure 4: How approximation accuracy improves over training updates for FastSVERL's explanation models across `Mastermind-222` (left), `Mastermind-333` (middle), and `Gridworld` (right). Each line shows the mean squared error (MSE) between predicted and exact values, averaged over all states and features. Shaded regions, which are negligible, indicate standard error over 20 runs. As you move down through the rows, downstream models (e.g. the behaviour Shapley model) use exact or approximate upstream models from earlier plots (e.g. the behaviour characteristic). Empty slots correspond to the outcome explanations that cannot feasibly be computed exactly in `Mastermind-333`.

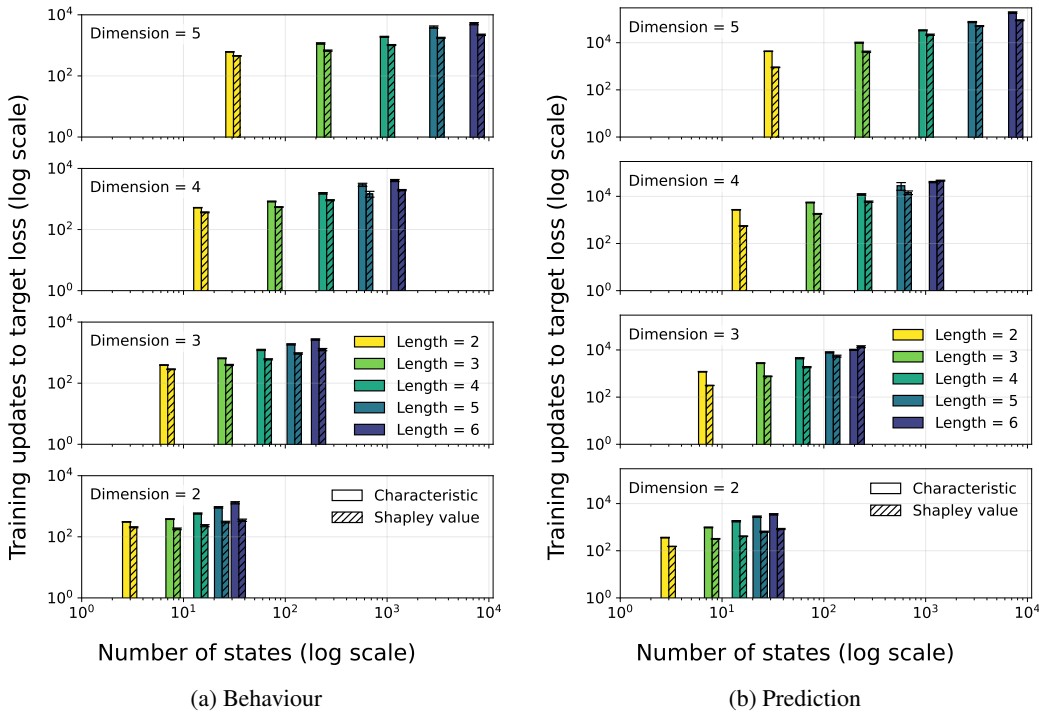

(a) Behaviour

(b) Prediction

Figure 5: Training batches required to reach a fixed target loss (0.01) when approximating characteristic and Shapley values for behaviour and prediction in `Hypercube`; standard error over 20 runs. Each subplot fixes the number of features $n$ (i.e. dimensions). Bar colour indicates cube side length $l$.

**Scalability.** We extend the illustration of FastSVERL's scalability presented in Figure 2 by applying it to prediction explanations in the `Hypercube` domain. Outcome explanations were not included because exact Shapley values were infeasible to compute for the larger cubes. Variability across runs is limited to randomness in training, with all experimental conditions fixed and the entire process fully seeded for controlled reproducibility. Figure 5 shows the training updates required to reach a fixed target loss (0.01) for both behaviour and prediction explanations. Consistent with the findings from behaviour explanations, the number of training updates scales approximately polynomially with the number of states, while remaining relatively insensitive to the number of features.

**Large-scale convergence results.** We now extend the large-scale domain analysis from Table 1 to the prediction and performance explanation models. As with the behaviour explanations, direct *accuracy* validation is infeasible in these domains. We therefore examine the same three key properties: (1) if the models converge, (2) the stability of that convergence, and (3) if the computational cost remains manageable as domain complexity grows.

Table 3 presents the convergence results for the prediction models. Consistent with the findings for the behaviour models, both the characteristic and Shapley models converge to a stable loss, and the number of training updates required remains consistent across the domains.

Next, Table 4 presents the results for the performance models. For the performance explanation, we present results only for the Shapley model. The associated characteristic model uses a bootstrapped reinforcement learning objective that does not converge to a fixed value and was instead trained for a fixed number of updates—the same number of updates required to train the agent. The Shapley models were trained using only the off-policy performance characteristic. The results again show stable and efficient convergence, consistent with the findings in the main body.

To complement the results in the tables above, Figure 6 visualises the full training curves for each model, illustrating the rates of convergence.

Table 3: Convergence of prediction models in large-scale `Mastermind` domains over 10 runs.

| Domain | Model | Updates to Converge (Mean $\pm$ Std. Err.) | Final Loss (Mean $\pm$ Std. Err.) |
|---|---|---|---|
| Mastermind-443 | Characteristic | $(1.08 \pm 0.06) \times 10^6$ | $(2.54 \pm 0.04) \times 10^{-1}$ |
| | Shapley | $(6.01 \pm 0.51) \times 10^5$ | $(1.57 \pm 0.07) \times 10^{-1}$ |
| Mastermind-453 | Characteristic | $(1.07 \pm 0.11) \times 10^6$ | $(2.74 \pm 0.04) \times 10^{-1}$ |
| | Shapley | $(5.93 \pm 0.46) \times 10^5$ | $(1.77 \pm 0.10) \times 10^{-1}$ |
| Mastermind-463 | Characteristic | $(9.88 \pm 0.34) \times 10^5$ | $(3.27 \pm 0.04) \times 10^{-2}$ |
| | Shapley | $(9.18 \pm 0.63) \times 10^5$ | $(1.70 \pm 0.06) \times 10^{-1}$ |

Table 4: Convergence of performance Shapley models in large-scale `Mastermind` domains over 10 runs.

| Domain | Model | Updates to Converge (Mean $\pm$ Std. Err.) | Final Loss (Mean $\pm$ Std. Err.) |
|---|---|---|---|
| Mastermind-443 | Shapley | $(6.79 \pm 0.63) \times 10^5$ | $(1.58 \pm 0.07) \times 10^{-1}$ |
| Mastermind-453 | Shapley | $(5.70 \pm 0.44) \times 10^5$ | $(6.33 \pm 0.30) \times 10^{-1}$ |
| Mastermind-463 | Shapley | $(5.56 \pm 0.34) \times 10^5$ | $(7.57 \pm 0.26) \times 10^{-1}$ |

**Full qualitative results.** We extend the qualitative illustration of the framework's behaviour insights in Table 2 by providing the full set of visual explanations in our project's code repository.[2] This extends the single illustrative example by presenting the behaviour, prediction, and performance explanations for all states encountered by an optimal policy in each of the large-scale `Mastermind` domains.

# D Off-policy learning for explanation models

This section presents the full derivation of the off-policy importance sampling approach introduced in Section 4, along with a complete set of empirical illustrations.

## D.1 Theory and derivation

FastSVERL trains characteristic and Shapley models using samples from the steady-state distribution $p^\pi(s)$ of the policy being explained. But what if the agent cannot interact with the environment to collect this data, for example, when further interaction is costly? In such settings, we propose sampling from a replay buffer $\mathcal{B}$ containing transitions gathered during training.

Because the buffer $\mathcal{B}$ aggregates data from a sequence of past policies $\{\pi_t\}_{t=0}^T$, its marginal state distribution $p^\mathcal{B}(s)$ differs from the target distribution $p^\pi(s)$. To correct the distributional mismatch, we apply importance sampling [18], reweighting each sample drawn from $\mathcal{B}$. For any loss of the form

$$\mathcal{L}(\beta) = \mathop{\mathbb{E}}_{s \sim p^\pi} [\ell(s; \beta)], \qquad (35)$$

we rewrite the expectation under $p^\pi(s)$ using samples from $p^\mathcal{B}(s)$:

$$\mathcal{L}(\beta) = \mathop{\mathbb{E}}_{s \sim p^\mathcal{B}} \left[ \frac{p^\pi(s)}{p^\mathcal{B}(s)} \cdot \ell(s; \beta) \right]. \qquad (36)$$

This yields an unbiased estimate of the original loss. However, neither $p^\pi(s)$ nor $p^\mathcal{B}(s)$ is known explicitly, so we approximate the density ratio using action probabilities. Since each transition $(s_t, a_t) \in \mathcal{B}$ was generated by a known behaviour policy $\pi_t$, we propose estimating the importance

---

[2]The complete results are available at: `https://github.com/djeb20/fastsverl`.

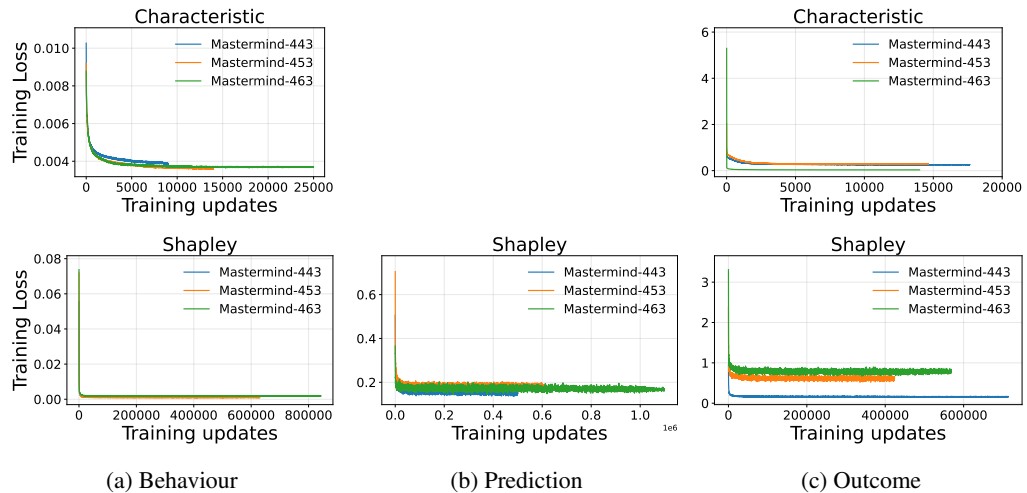

|  | | |
|---|---|---|
| (a) Behaviour | (b) Prediction | (c) Outcome |

Figure 6: Training loss curves for the characteristic (top row) and Shapley (bottom row) models in the large-scale `Mastermind` domains. Each line represents the mean loss over training updates, averaged across all three domain sizes (`Mastermind-443`, `Mastermind-453`, and `Mastermind-463`). Shaded regions indicate standard error over 10 runs. The empty slot corresponds to the performance characteristic model.

weight as:

$$\mathcal{L}(\beta) \approx \mathop{\mathbb{E}}_{(s_t, a_t) \sim \mathcal{B}} \left[ \frac{\pi(s_t, a_t)}{\pi_t(s_t, a_t)} \cdot \ell(s_t; \beta) \right]. \tag{37}$$

The reweighting in Equation 37 estimates the loss by weighting each sample in proportion to its relevance under $\pi$.

Two practical factors may affect the stability and accuracy of the importance-weighted loss in Equation 37. The first is the similarity between the target policy $\pi$ and the past policies $\{\pi_t\}$ that generated the buffer. The more these policies differ, the higher the variance of the importance weights, increasing the variance of the loss estimate. The second is the buffer's coverage of the state distribution $p^\pi(s)$. If states commonly visited by $\pi$ are underrepresented, their corresponding loss terms may be poorly estimated. In practice, transitions from later-stage policies are more likely to resemble the final policy $\pi$, so additional interaction near the end of training may improve both policy similarity and state coverage in the buffer.

When importance weights exhibit high variance, techniques such as weight clipping [24] or adaptive weighting [30] can help stabilise training. We illustrate these strategies empirically in the next section.

## D.2 Empirical illustrations

We present three empirical illustrations of off-policy training in FastSVERL. First, we extend the experiment from Figure 3a by training prediction characteristic models off-policy in `Mastermind-222`. We then investigate the impact of two variance-control strategies: adaptive weighting [30] and weight clipping [24]. All figures report approximation accuracy over training updates, measured by mean squared error (MSE) between predicted and exact values, averaged across states and features. Shaded regions indicate the standard error over 20 runs. Variability across runs is limited to randomness in the training pipeline, with all experimental conditions fixed and the entire process fully seeded for controlled reproducibility. Standard errors are corrected to account for this variability [16].

**Learning characteristics off-policy in `Mastermind-222`.** Figure 7 extends the illustration of off-policy training from Figure 3a to prediction. Consistent with the findings in Section 4: (1) importance sampling reduces approximation error relative to using the training buffer without reweighting, suggesting that correcting for distributional mismatch improves performance; and (2) approximation error remains higher than under on-policy training, indicating that even with reweighting, off-policy data is an imperfect substitute.

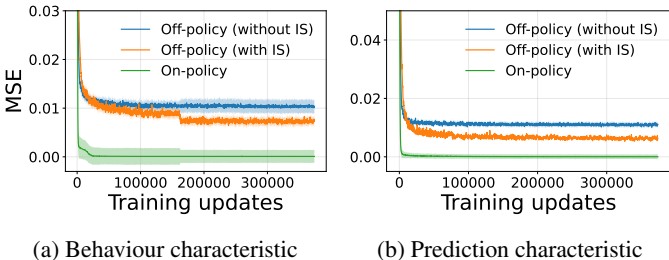

(a) Behaviour characteristic  (b) Prediction characteristic

Figure 7: Approximation accuracy of off-policy training in `Mastermind-222` with models trained using either on-policy data, or off-policy data with two configurations: (1) without importance sampling (IS) and (2) with IS.

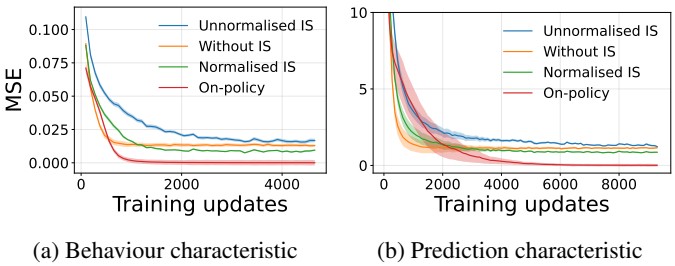

(a) Behaviour characteristic  (b) Prediction characteristic

Figure 8: Approximation accuracy of off-policy training in `Gridworld` with models trained using either on-policy data, or off-policy data with three configurations: (1) without importance sampling (IS), (2) with unnormalised IS, and (3) with normalised IS.

**Normalising importance weights.**   A common strategy to reduce variance in importance sampling is to normalise weights within each batch by dividing them by their sum [30]. We illustrate this strategy by training behaviour and prediction characteristic models in `Gridworld` under three off-policy conditions: (1) without importance sampling, (2) with unnormalised importance sampling, and (3) with normalised importance sampling. An on-policy configuration is included as a baseline. The results, shown in Figure 8, suggest that normalising the importance weights is crucial: using unnormalised weights leads to worse approximation accuracy than no reweighting at all, while normalised weights yield the best results. We therefore apply weight normalisation in all experiments using off-policy sampling.

**Clipping importance weights.**   Another common strategy to reduce variance in importance sampling is to clip the weights to a fixed range [24]. We illustrate this strategy by training behaviour and prediction characteristic models in `Gridworld` under several off-policy conditions: (1) without importance sampling, (2) with unclipped importance sampling, and (3) with importance weights clipped symmetrically around 1 to the range $[1-c, 1+c]$. We consider thresholds of $c = 0.99, 0.995$, and $0.998$, selected to yield a spread of performance curves. An on-policy configuration is included as a baseline. The results, shown in Figure 9, indicate that reducing variance through clipping may impair performance: unclipped importance sampling achieves the highest accuracy, while tighter clipping progressively harms approximation. The extreme case of $c = 0$, equivalent to no importance sampling, performs the worst. These results suggest a bias-variance trade-off in importance sampling, where retaining the full range of weights avoids the bias introduced by clipping and may be preferred to mitigating variance in this setting.

## E   Empirical illustrations of continuously learning to explain

We extend the illustration of jointly training a DQN agent and explanation models presented in Figure 3b to all explanation types in the domains `Mastermind-222` and `Gridworld`. Variability across runs is limited to randomness in the training pipeline, with all experimental conditions fixed and the entire process fully seeded for controlled reproducibility.

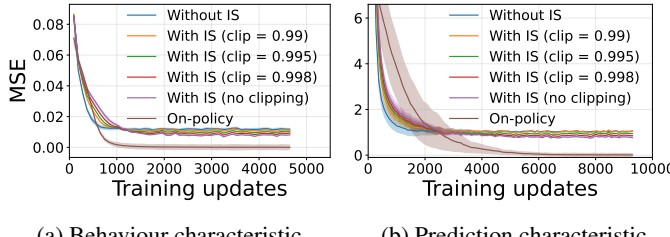

(a) Behaviour characteristic       (b) Prediction characteristic

Figure 9: Approximation accuracy of off-policy training in `Gridworld` with models trained using either on-policy data, or off-policy data with three configurations: (1) without importance sampling (IS), (2) with unclipped IS, and (3) with IS clipped at thresholds of 0.99, 0.995, and 0.998.

Figure 10 presents the approximation accuracy for the Shapley models across varying update ratios (1:1, 2:1, 10:1, and 50:1), alongside the agent's expected return and DQN loss to track the impact of policy changes during training. Consistent with the findings in Section 4, increasing the update ratio mitigates error spikes during rapid policy shifts, reflecting better alignment between the agent and the explanation models. That is, apart from the behaviour explanation for `Gridworld`, where varying the update rate has minimal effect on error reduction. This may be due to the already low error magnitudes, suggesting the bottleneck could instead be linked to another source of error, such as the off-policy nature of the data.

## F    Sampling-based approximation of characteristic functions

Training characteristic models is a major computational bottleneck in FastSVERL. In Section 5, we proposed removing this cost by integrating cheap single-sample approximations of characteristic values into the Shapley model loss, demonstrating this for the behaviour characteristic. The same approach can be applied to the prediction characteristic, which also admits a natural sampling-based approximation.

In this section, we complete the proposal by applying it to all explanation models that rely on either the behaviour or prediction characteristic. We first consider their use in training Shapley models, then turn to sampling the behaviour characteristic in training the outcome characteristic. We conclude with the full set of empirical illustrations of the proposals.

**Behaviour Shapley.**    We train a parametric model $\hat{\phi}(s, a; \theta)$ to predict the Shapley contributions of each feature of state $s$ to the agent's probability of selecting action $a$. Instead of querying a characteristic model, we replace the characteristic function $\tilde{\pi}_s^a(\mathcal{C})$ in the Shapley model loss with a single-sample approximation. At each loss evaluation, a state $s' \sim p^\pi(\cdot \mid s^{\mathcal{C}})$ is sampled from the conditional steady-state distribution, and the characteristic value is approximated using $\pi(s', a)$. The model is trained to minimise the following loss:

$$\mathcal{L}(\theta) = \mathop{\mathbb{E}}_{p^\pi(s)} \mathop{\mathbb{E}}_{\mathrm{Unif}(a)} \mathop{\mathbb{E}}_{p(\mathcal{C})} \mathop{\mathbb{E}}_{s' \sim p^\pi(\cdot \mid s^{\mathcal{C}})} \left| \pi(s', a) - \tilde{\pi}_s^a(\emptyset) - \sum_{i \in \mathcal{C}} \hat{\phi}^i(s, a; \theta) \right|^2 . \tag{38}$$

After training, the Shapley model's output is corrected to satisfy the efficiency constraint:

$$\phi^i(\tilde{\pi}_s^a) \approx \hat{\phi}^i(s, a; \theta) + \frac{1}{|\mathcal{F}|} \left( \pi(s, a) - \tilde{\pi}_s^a(\emptyset) - \sum_{j \in \mathcal{F}} \hat{\phi}^j(s, a; \theta) \right) . \tag{39}$$

With a sufficiently expressive model class and exact optimisation, the corrected output of the global optimum $\hat{\phi}(s, a; \theta^*)$ recovers exact Shapley values for all state-action pairs $(s, a)$ such that $p^\pi(s) > 0$ and $a \in \mathcal{A}$.

In practice, we approximate the conditional steady-state distribution $p^\pi(s' \mid s^{\mathcal{C}})$ by sampling from a replay buffer containing transitions collected under $\pi$, uniformly selecting states $s'$ such that $s'^{\mathcal{C}} = s^{\mathcal{C}}$.

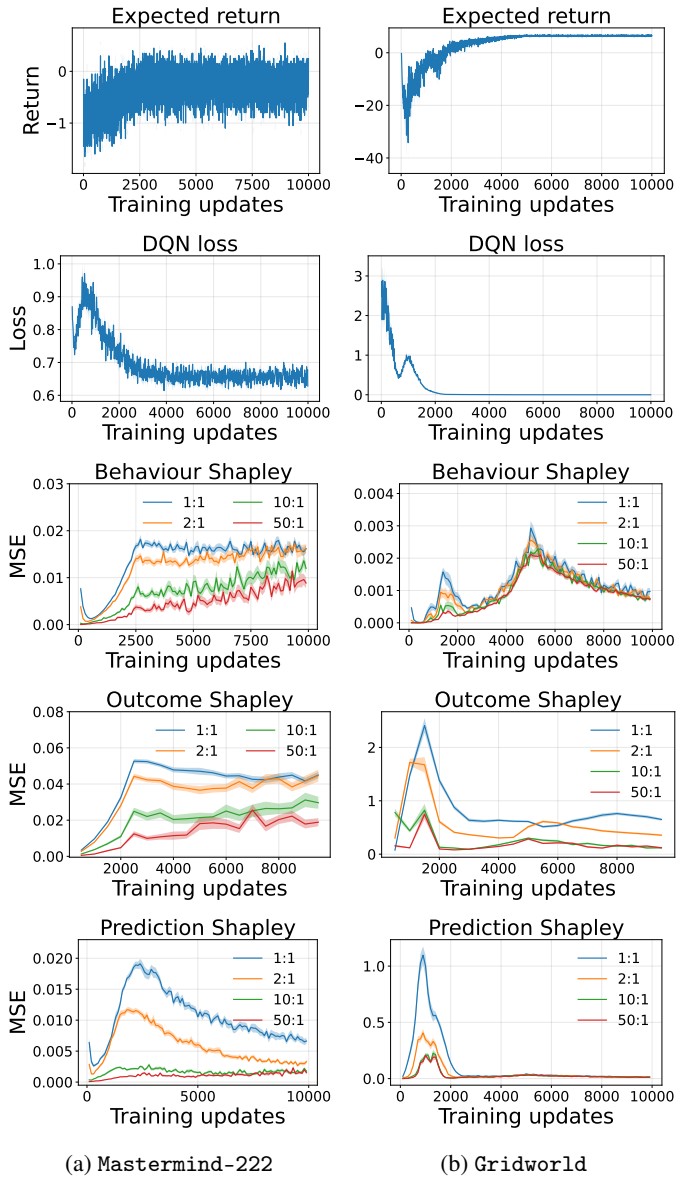

(a) `Mastermind-222`  (b) `Gridworld`

Figure 10: Approximation accuracy of Shapley values trained in parallel with the agent under explanation-to-agent update ratios of 1:1, 2:1, 10:1, and 50:1 in `Mastermind-222` (left) and `Gridworld` (right). Each line represents the mean squared error (MSE) between predicted and exact values, averaged over all states and features. Shaded regions indicate the standard error over 20 runs. The agent's expected return and DQN loss are also shown to illustrate learning dynamics during training.

Alternatively, one could train a parametric model of this distribution [8], which may generalise better to sparsely sampled regions but reintroduces the cost of learning an additional model.

The null characteristic value $\tilde{\pi}_s^a(\emptyset)$ is similarly estimated via Monte Carlo sampling from the steady-state distribution:

$$\tilde{\pi}_s^a(\emptyset) = \mathop{\mathbb{E}}_{s \sim p^\pi(s)} \left[ \pi(s, a) \right]. \tag{40}$$

This estimate is computationally efficient: it can be computed once by passing the entire replay buffer through the policy network in parallel and averaging the results, then reused across all loss evaluations.

**Prediction Shapley.** We train a parametric model $\hat{\phi}(s; \theta) : \mathcal{S} \rightarrow \mathbb{R}^{|\mathcal{F}|}$ to predict the Shapley contributions of each feature of state $s$ to the agent's value estimate $\hat{v}^\pi(s)$. We replace the prediction characteristic $\hat{v}_s^\pi(\mathcal{C})$ in the Shapley model loss with a single-sample approximation. At each loss evaluation, a state $s' \sim p^\pi(\cdot \mid s^\mathcal{C})$ is sampled from the conditional steady-state distribution, and the characteristic value is approximated using $\hat{v}^\pi(s')$. The model is trained to minimise the following loss:

$$\mathcal{L}(\theta) = \mathop{\mathbb{E}}_{p^\pi(s)} \mathop{\mathbb{E}}_{p(\mathcal{C})} \mathop{\mathbb{E}}_{s' \sim p^\pi(\cdot \mid s^\mathcal{C})} \left| \hat{v}^\pi(s') - \hat{v}_s^\pi(\emptyset) - \sum_{i \in \mathcal{C}} \hat{\phi}^i(s; \theta) \right|^2 . \tag{41}$$

After training, the model output is corrected to satisfy the efficiency constraint:

$$\phi^i(\hat{v}_s^\pi) \approx \hat{\phi}^i(s; \theta) + \frac{1}{|\mathcal{F}|} \left( \hat{v}^\pi(s) - \hat{v}_s^\pi(\emptyset) - \sum_{j \in \mathcal{F}} \hat{\phi}^j(s; \theta) \right) . \tag{42}$$

With a sufficiently expressive model class and exact optimisation, the corrected output of the global optimum $\hat{\phi}(s; \theta^*)$ recovers the exact Shapley values for all $(s, \mathcal{C})$ such that $p^\pi(s) > 0$ and $p(\mathcal{C}) > 0$.

**Behaviour convergence proof.** We now prove that the learning objective in Equation 38 recovers exact Shapley values at the global optimum. This result generalises to the prediction objective in Equation 41 because they share the same structure.

We make the following assumptions:

1. The model $\hat{\phi}(s, a; \theta)$ is selected from a function class expressive enough to represent the true Shapley value function $\phi(\tilde{\pi}_s^a)$ for all state-action pairs $(s, a)$ such that $p^\pi(s) > 0$.

2. The global minimum of the loss $\mathcal{L}(\theta)$ exists and is attained.

3. States $s$ are sampled from the steady-state distribution $p^\pi(s)$, actions from the uniform distribution over $\mathcal{A}$, subsets $\mathcal{C}$ from a fixed distribution over all subsets of $\mathcal{F}$, and samples $s' \sim p^\pi(\cdot \mid s^\mathcal{C})$ from the conditional steady-state distribution.

For a fixed state-action pair $(s, a)$ such that $p^\pi(s) > 0$, the sampling-based loss in Equation 38 reduces to:

$$\mathop{\mathbb{E}}_{p(\mathcal{C})} \mathop{\mathbb{E}}_{s' \sim p^\pi(\cdot \mid s^\mathcal{C})} \left| \pi(s', a) - \tilde{\pi}_s^a(\emptyset) - \sum_{i \in \mathcal{C}} \hat{\phi}^i(s, a; \theta) \right|^2 . \tag{43}$$

Since $\pi(s', a)$ is the only term that depends on $s'$, we may equivalently write:

$$\mathop{\mathbb{E}}_{p(\mathcal{C})} \left| \mathop{\mathbb{E}}_{s' \sim p^\pi(\cdot \mid s^\mathcal{C})} [\pi(s', a)] - \tilde{\pi}_s^a(\emptyset) - \sum_{i \in \mathcal{C}} \hat{\phi}^i(s, a; \theta) \right|^2 . \tag{44}$$

By the definition of the behaviour characteristic function, we have:

$$\mathop{\mathbb{E}}_{p(\mathcal{C})} \left| \tilde{\pi}_s^a(\mathcal{C}) - \tilde{\pi}_s^a(\emptyset) - \sum_{i \in \mathcal{C}} \hat{\phi}^i(s, a; \theta) \right|^2 . \tag{45}$$

This expression is identical to the reduced Shapley model loss in Equation 25, bringing us to the same point as in the proof presented in Appendix A. The remainder of the proof proceeds identically. By enforcing the efficiency constraint via an additive correction [22],

$$\phi^i(\tilde{\pi}_s^a) := \hat{\phi}^i(s, a; \theta) + \frac{1}{|\mathcal{F}|} \left( \pi(s, a) - \tilde{\pi}_s^a(\emptyset) - \sum_{j \in \mathcal{F}} \hat{\phi}^j(s, a; \theta) \right) , \tag{46}$$

the unique minimiser of the loss is given by the Shapley values $\phi^i(\tilde{\pi}_s^a)$ [5]. Therefore, the globally optimal parameters $\theta^*$ recover exact Shapley values for all $(s, a)$ such that $p^\pi(s) > 0$.

∎

**Remark.** The sampling-based approximations of the behaviour and prediction Shapley values are unbiased because their objectives recover the original asymptotically unbiased model-based losses in expectation.

**Outcome characteristic.** The outcome characteristic function $\tilde{v}_s^\pi(\mathcal{C})$ is defined as the expected return received from state $s$ when the agent's policy has access only to the features in $\mathcal{C}$:

$$\tilde{v}_s^\pi(\mathcal{C}) := \mathbb{E}_\mu\left[G_t \mid S_t = s\right], \tag{47}$$

where $\mu$ is a modified policy that selects actions using the behaviour characteristic function $\tilde{\pi}_s^a(\mathcal{C})$ when features are unknown in state $s$, and follows the original policy $\pi$ elsewhere.

To estimate $\tilde{v}_s^\pi(\mathcal{C})$, FastSVERL defines a conditioned policy $\hat{\pi}(a \mid s; s_e, \mathcal{C})$ that follows the behaviour characteristic model $\hat{\pi}(s, a \mid \mathcal{C}; \beta)$ when $s = s_e$, and follows the original policy $\pi$ elsewhere. A parametric value function $V(s \mid s_e, \mathcal{C}; \beta)$ is then trained to predict the expected return under this conditioned policy for each $(s_e, \mathcal{C})$ pair.

Training this outcome characteristic model requires access to a pre-trained behaviour characteristic model, introducing additional computational cost. To avoid this, we instead bypass the behaviour characteristic model entirely by sampling a state $s' \sim p^\pi(\cdot \mid s_e^\mathcal{C})$ and using $\pi(s', a)$ as the action probability whenever the conditioned policy $\hat{\pi}(a \mid s; s_e, \mathcal{C})$ would otherwise act according to the behaviour characteristic model—that is, when $s = s_e$.

The proposed sampling approach applies only to the on-policy formulation of the outcome characteristic. In this setting, the value function $V(s \mid s_e, \mathcal{C}; \beta)$ is trained using a standard bootstrapped DQN-style loss:

$$\mathcal{L}(\beta) = \mathbb{E}_{(s,r,s',s_e,\mathcal{C})\sim\mathcal{B}} \left|r + \gamma V'(s' \mid s_e, \mathcal{C}; \beta^-) - V(s \mid s_e, \mathcal{C}; \beta)\right|^2, \tag{48}$$

where the buffer $\mathcal{B}$ contains transitions collected from the conditioned policy when sampling is used to approximate the behaviour characteristic at $s = s_e$.

An off-policy variant was also considered in Section 3.2, where a parametric state-action value function $Q(s, a \mid s_e, \mathcal{C}; \beta)$ is trained and the outcome characteristic is recovered as

$$\tilde{v}_s^\pi(\mathcal{C}) \approx \sum_{a\in\mathcal{A}} \hat{\pi}(s, a \mid s_e, \mathcal{C}) \cdot Q(s, a \mid s_e, \mathcal{C}; \beta). \tag{49}$$

Since this recovery step requires querying the behaviour characteristic model, which is no longer available under the sampling-based approximation, the method cannot be applied in this setting.

**Empirical illustrations.** We extend the analysis of replacing characteristic models with sampling, as presented in Figure 3c, to prediction Shapley models and on-policy outcome characteristic models across `Mastermind-222`, `Mastermind-333`, and `Gridworld`. Variability across runs is limited to randomness in the training process, with all experimental conditions fixed and the entire pipeline fully seeded. Explanation models are trained using a single agent instance across all runs.

Figure 11 shows the explanation model's losses over cumulative training updates. Consistent with the findings in Section 5, sampling-based approximations converge faster than their model-based counterparts and eliminate error propagation from characteristic models. However, minimal computational gains are observed for the outcome characteristic, as the cost of training the behaviour characteristic is relatively small compared to the substantial cost of training the outcome characteristic itself. This suggests that while sampling effectively reduces error propagation, its impact on computational efficiency for outcome explanations is limited.

# G   Domains

This section provides descriptions of the reinforcement learning domains used in the experiments.

**Gridworld** [2], is a deterministic $2 \times 4$ environment where the agent's state is defined by its $(x, y)$ grid coordinates. The state space is:

$$\mathcal{S} = \{(1,1), (1,3), (1,4), (2,1), (2,2), (2,3), (2,4)\}. \tag{50}$$

Each episode begins in a start state sampled uniformly from $(1, 1)$ and $(2, 1)$. The agent can take actions `North`, `East`, `South`, and `West`; actions that would move the agent outside the grid or into

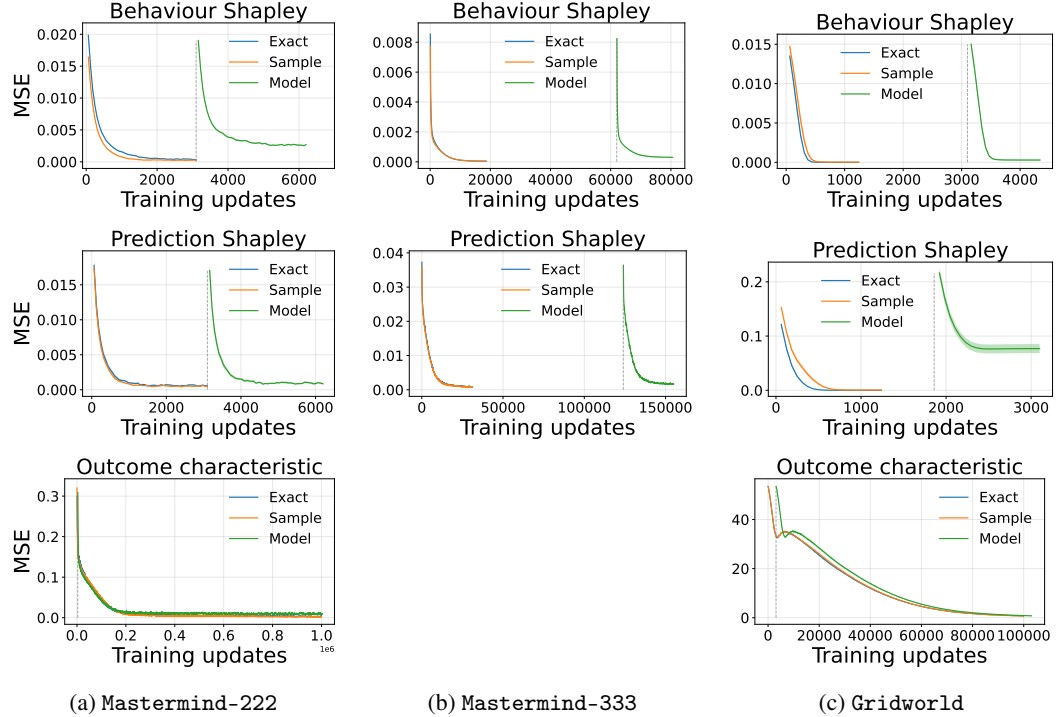

|(a) `Mastermind-222`|(b) `Mastermind-333`|(c) `Gridworld`|

Figure 11: Approximation accuracy of Shapley and characteristic values trained with sampled, exact or model-based characteristics in `Mastermind-222` (left), `Mastermind-333` (middle), and `Gridworld` (right). The empty slot corresponds to the outcome characteristic that cannot feasibly be computed exactly in `Mastermind-333`. Each line represents the mean squared error (MSE) between predicted and exact values, averaged over all states and features. Shaded regions indicate standard error over 20 runs. The dashed lines mark the end of pre-training for the characteristic models in the model-based approach.

the missing state $(1, 2)$ are treated as invalid, incurring a reward but leaving the agent's position unchanged.

The terminal states are $(1, 4)$ and $(2, 4)$. The agent receives a reward of $-1$ per decision stage and an additional $+10$ upon reaching a terminal state, making this a shortest-path problem. The optimal policy moves `East` in $(1, 1)$ and `North` in all other states to minimise the number of steps to termination.

**Mastermind** [2] is a reinforcement learning adaptation of the classic code-breaking game. At the start of each episode, the environment randomly samples a hidden code consisting of a sequence of letters. The agent must identify the code within a fixed number of guesses. After each guess, the environment returns two types of feedback:

- **Position clue:** the number of letters that are both correct and in the correct position.
- **Misplaced clue:** the number of correct letters that are in the wrong position.

These values are computed sequentially: letters that are correct and in the correct position contribute to the position clue and are excluded when computing the misplaced clue, which counts only the remaining correct letters in the wrong position.

Each state encodes the agent's current game board: a sequence of previous guesses and the corresponding feedback. States are feature-based, with each guess represented by a fixed number of features: one per letter in the guess, plus two features for the position and misplaced clues. Unused guesses are represented using a dedicated `empty` value.

The environment can be configured by varying the code length, the number of guesses, and the size of the letter set (i.e. the alphabet). We consider five configurations:

- `Mastermind-222`: Codes of length 2 drawn from the alphabet $\{A, B\}$, yielding 4 possible codes. The agent is allowed up to 2 guesses. The state space contains 53 unique states, each represented by 8 features. Each code corresponds to a unique action, giving 4 available actions.

- `Mastermind-333`: Codes of length 3 drawn from the alphabet $\{A, B, C\}$, yielding 27 possible codes. The agent is allowed up to 3 guesses. The state space contains over 100,000 states, each with 15 features. There are 27 available actions.

- `Mastermind-443`: Codes of length 4 drawn from the alphabet $\{A, B, C\}$, yielding 81 possible codes. The agent is allowed up to 4 guesses. The state space contains over $4.3 \times 10^7$ states, each with 24 features. There are 81 available actions.

- `Mastermind-453`: Codes of length 4 drawn from the alphabet $\{A, B, C\}$, yielding 81 possible codes. The agent is allowed up to 5 guesses. The state space contains over $3.5 \times 10^9$ states, each with 30 features. There are 81 available actions.

- `Mastermind-463`: Codes of length 4 drawn from the alphabet $\{A, B, C\}$, yielding 81 possible codes. The agent is allowed up to 6 guesses. The state space contains over $2.8 \times 10^{11}$ states, each with 36 features. There are 81 available actions.

The reward function assigns $-1$ per guess and provides an additional reward equal to the maximum number of guesses if the agent correctly identifies the hidden code. Episodes terminate when the correct code is guessed or the guess limit is reached.

## H   Experimental setup and compute resources

All experimental configurations, including hyperparameters, training settings, and environment details, are included in the project code's test scripts.[3] These scripts are designed to produce fully reproducible and identical results for every experiment. The DQN agent used throughout the experiments is based on the implementation from CleanRL [11].

The hyperparameters for all agents, characteristic models, and Shapley models were pragmatically chosen without tuning, as the experiments are intended to illustrate FastSVERL's properties rather than benchmark against alternative methods. Initial values were selected, found to be sufficient for learning, and kept constant across experiments unless they were the specific subject of study or directly linked to design choices being evaluated. The only exception to this approach was the choice of the masking value used in behaviour and prediction characteristic models to represent unknown features. Although the theoretical framework permits any value outside the support of $\mathcal{S}$, we found that large magnitude values hindered training stability, possibly due to amplified gradient magnitudes. In contrast, smaller magnitude values, closer to the support of $\mathcal{S}$, resulted in smoother learning and were adopted for all experiments.

Standard errors in all experiments are calculated using the standard error of the mean, corresponding to 1-sigma error bars and shaded areas. To better isolate variability arising from the experimental conditions, and not noise introduced by unrelated stochastic factors, we sometimes apply the correction method proposed by Masson and Loftus [16], which adjusts for run-specific differences that are unrelated to the treatment effect. Specific sources of variability, such as agent initialisation, are detailed alongside each experimental setup in the appendix sections.

All experiments were conducted on a local workstation equipped with the following specifications:

- Processor: Intel i9-14900K (24 cores, up to 6.0GHz)
- GPU: NVIDIA RTX 4090 (24GB VRAM)
- Memory: 96GB DDR5 RAM
- Storage: 2x1TB NVMe (Samsung 990 EVO) and 4TB SSD (Samsung 870 QVO)

The full research project required substantially more compute than the experiments reported here, including preliminary testing and exploratory experiments not included in the final results.

---

[3]FastSVERL code is available at: `https://github.com/djeb20/fastsverl`.

