# OpenReview forum: "Approximating Shapley Explanations in Reinforcement Learning"
_NeurIPS.cc/2025/Conference — NeurIPS 2025 poster_

### Official Review · Reviewer_qBRx · 2025-06-22

**Clarity:** 3
**Significance:** 3
**Originality:** 2
**Rating:** 3
**Confidence:** 3

**Summary:**

The paper studies Shapley value explanations in RL and proposes a new efficient approximation method. Exactly computing SVs is known to be a computationally costly procedure in high-dimensional domains hindering practical scalability. The problem has been extensively addressed in the SL domain. However, there is no method tailored to the unique challenges the RL framework poses, such as temporal dependencies and coverage issues when only offline/off-policy data are available. The authors propose FastSVERL: a parametric approach for approximating state feature contributions (to policy action probs; expected and realized values) via SV in an efficient way.

**Questions:**

It was not clear to me how temporal dependencies, characteristic of RL environments, were explictly addressed by your framework.

**Ethical Concerns:**

["NO or VERY MINOR ethics concerns only"]

**Final Justification:**

The changes in the experiments although strengthening the paper constitute a change too big to be done at this point of the revewing process

**Limitations:**

Mentioned

**Paper Formatting Concerns:**

All good

**Quality:**

2

**Strengths And Weaknesses:**

**Strengths**

- The paper provides a comprehensive summary of SVERL [1,2], allowing readers unfamiliar with this work to easily follow along. In general, the paper is very well-written and easy to read.
- The problem addressed is nicely posed and well motivated.
- I found the proposed approach to be reasonable, albeit most components are derived from existing work. I found the single conditioned policy idea (line 203) a nice touch.


**Weaknesses**

- The main weak point of this paper is the limited empirical evaluation (main reason for giving a low score). I would typically expect from such a paper to conduct extensive experiments in more complex ("real-world") environments. Furthermore, since the authors chose to use simpler simulated benchmarks, where ground truth attributions can be computed, the use of more baseline methods for comparison are needed. Such baseline methods would include SL techniques mentioned in the background section.
- It was not clear from the write-up how some of the challenges recognized as unique to the RL method were explicitly addressed by the FastSVERL approach (see my questions below).
- [Minor] since there are works in (MA)RL using SV for different purposes than the one considered in the paper, it would good to specify ealrier in the paper (even from appendix) the use-case of focus.

---

> ### Author Rebuttal · Authors · 2025-07-29
>
> Thank you for your positive and encouraging feedback, particularly on the paper's clarity. We are happy to address the weaknesses and questions raised in the review below.
>
> **On the Originality of Contributions (Strength 3)**
>
> We appreciate that the proposed framework builds on established concepts from SVERL and the supervised learning literature. However, adapting these concepts to the practical reinforcement learning setting was a non-trivial process that required several contributions; a direct application of existing methods is insufficient for the unique challenges that reinforcement learning presents, which we discuss in more detail later.
>
> For example, while the approximation of the performance characteristic involves the familiar process of learning a value function, its application to a conditioned policy (as described in Section 3.2) was a considered approach necessary to make the problem computationally tractable in the Shapley value setting. Similarly, to the best of our knowledge, this is the first work to address approximating Shapley values using off-policy data or for non-stationary models. These are core reinforcement learning challenges of practical interest to the community; extending existing Shapley value methods to these dynamic settings required careful consideration and was not a direct application.
>
> Furthermore, we introduce a more efficient approximation method altogether. The proposed sampling-based update (Section 5) is a significant algorithmic contribution that improves both computational efficiency and accuracy by replacing the need for a characteristic model entirely. This contribution is general and also directly applicable to improving Shapley value approximations in supervised learning, demonstrating originality beyond the specific context of reinforcement learning.
>
> **On Limited Evaluation in Complex Environments (Weakness 1)**
>
> Thank you for this critical point about evaluating the proposed methods in more complex environments. We agree that the real-world applicability of a method is essential.
>
> Our deliberate methodological use of smaller domains was a direct consequence of the need to compute exact Shapley values for validation. Approximation accuracy can be rigorously tested *only* in these verifiable settings. Therefore, our experiments were necessarily constrained to this regime to ensure scientific validity.
>
> It is only within this verifiable regime that we can demonstrate how the proposed framework’s accuracy scales. Our Hypercube experiments were designed specifically for this purpose, and they showed that the computational cost of achieving accurate explanations scales well with both the number of states and features. This graceful scaling of accuracy, in the one region where it can be principally measured, suggests the method should continue to perform well in larger domains. Furthermore, the results on Mastermind-333 in our supplement—with over 100,000 states and 15 features ($2^{15}$ subsets)—already represent a complex setting for Shapley value computation, providing further evidence of the framework's capabilities.
>
> That said, while we cannot directly assess accuracy in larger domains, we agree that general scalability deserves greater focus. Motivated by your feedback and that of others, we have run new experiments on substantially larger versions of Mastermind. We continue to use Mastermind specifically to allow fine-grained control over complexity. These new domains, the largest with 36 features and over 280 billion states, are designed to directly address the need for evaluation in more complex settings, particularly from a Shapley approximation perspective where the summation is over $2^{36}$ feature subsets.
>
> While it is impossible to measure accuracy against an exact ground truth in these domains, our theoretical guarantees and the strong accuracy demonstrated in verifiable settings suggest that FastSVERL's models will converge towards the correct values. Therefore, our goal with these new experiments is to test three key properties at scale: (1) if the models indeed converge, (2) the stability of that convergence across runs (measured by standard error), and (3) if the computational cost of convergence remains manageable.
>
> Due to space constraints, we report the mean updates to convergence and final training loss over 10 runs for a representative subset of results below. It is important to clarify that this loss is not a measure of ground-truth accuracy. The characteristic model's loss is designed to converge to a non-zero value as it learns to predict an expectation over the unobserved features. In contrast, the Shapley model's loss is designed to converge to zero, but this indicates it has successfully learned to explain the values provided by the characteristic model, not necessarily the ground truth.
>
> **Table 1: Convergence and Stability of Behaviour Explanations**
> | Domain | Explanation Model | Updates to Converge (Mean ± Std. Err.) | Final Loss (Mean ± Std. Err.) |
> | :--- | :--- | :---: | :---: |
> | **Mastermind-443** | Behaviour Characteristic | $(1.13 \pm 0.177) \times 10^6$ | $0.0038 \pm 0.000018$ |
> | (24 features, $\ge 4.3 \times 10^7$ states) | Behaviour Shapley | $(7.99 \pm 0.902) \times 10^5$ | $0.0013 \pm 0.000037$ |
> | **Mastermind-453** | Behaviour Characteristic | $(1.30 \pm 0.150) \times 10^6$ | $0.0036 \pm 0.000021$ |
> | (30 features, $\ge 3.5 \times 10^9$ states) | Behaviour Shapley | $(7.91 \pm 0.503) \times 10^5$ | $0.0010 \pm 0.000032$ |
> | **Mastermind-463** | Behaviour Characteristic | $(1.25 \pm 0.174) \times 10^6$ | $0.0037 \pm 0.000013$ |
> | (36 features, $\ge 2.8 \times 10^{11}$ states) | Behaviour Shapley | $(6.56 \pm 0.816) \times 10^5$ | $0.0019 \pm 0.000044$ |
>
> The results in Table 1 confirm that the models successfully converge in these large-scale domains. The standard error across multiple runs demonstrates that this convergence is stable and highly reliable. Most importantly, the number of training updates required for convergence remains similar even as the domain complexity grows.
>
> We also demonstrate that the significant computational advantages of the sampling-based approximations we propose (Section 5) hold at scale. This method is a key contribution of our work, and Table 2 confirms its efficiency benefits in these large domains.
>
> **Table 2: Total Computational Cost for Behaviour Explanations**
> | Domain | Total Updates (Model-Based) (Mean ± Std. Err.) | Total Updates (Sampling-Based) (Mean ± Std. Err.) |
> | :--- | :---: | :---: |
> | **Mastermind-443** | $(1.84 \pm 0.136) \times 10^6$ | $(5.69 \pm 0.498) \times 10^5$ |
> | **Mastermind-453** | $(2.09 \pm 0.095) \times 10^6$ | $(7.25 \pm 0.638) \times 10^5$ |
> | **Mastermind-463** | $(1.91 \pm 0.104) \times 10^6$ | $(6.06 \pm 0.584) \times 10^5$ |
>
> By eliminating the need to pre-train a characteristic model, the sampling-based approach reduces the total computational cost by over $2\times$ across all large-scale domains tested. This suggests that the substantial efficiency gains are consistent.
>
> We hope these clarifications and new results will help to address your concerns about empirical evaluation in more complex environments.
>
> **On the Lack of Baseline Methods (Weakness 1)**
>
> Thank you for the point about comparing to the Monte Carlo sampling methods mentioned in our background section.
>
> In a highly idealised setting (a fixed policy and resettable environment simulator) these methods could be adapted as a baseline. However, our work was motivated by the need to explain agents in more practical reinforcement learning scenarios.
>
> For the core challenge of explaining a non-stationary policy, Monte Carlo methods are computationally prohibitive because they must recompute all explanations from scratch for every policy update. Our parametric approach was chosen specifically to handle this dynamic setting because its parameters can be continually updated as the policy evolves.
>
> Furthermore, approximating the performance characteristic via sampling, as proposed in the original SVERL framework [1], requires multi-step rollouts from arbitrary start states. This approach depends on the often unreasonable assumption of a resettable simulator, which we deliberately avoided to create a more general and practical framework.
>
> **On Addressing Temporal Dependencies (Weakness 2, Q1)**
>
> Thank you, this question allows us to clarify a central aspect of FastSVERL. Temporal dependencies are addressed explicitly by the proposed approach to approximating performance explanations.
>
> The performance characteristic function is defined as the expected discounted sum of future rewards, a value inherently dependent on the entire subsequent trajectory. The influence of observing a subset of features in a state is therefore not an instantaneous effect. When features are unknown in a given state $s$, the agent follows a modified policy at that state for the entire episode. This alters the distribution over all future trajectories, and if the agent returns to state $s$ at a later time, it continues to follow this modified behaviour, compounding the temporal impact.
>
> The proposed approximation method—the conditioned policy and value function—was designed precisely to estimate this temporally-extended quantity in a tractable way.
>
> **On Specifying the Use-Case (Weakness 3)**
>
> Thank you for this helpful suggestion. We agree that explicitly distinguishing our use from other applications of Shapley values in reinforcement learning earlier in the paper would improve clarity for the reader.
>
> We thank you again for your time and the detailed and constructive feedback. We hope that our response and the new large-scale experiments have helped to address your initial concerns. We will incorporate the full set of these new results, along with all the clarifications discussed, into the final version of the manuscript. We hope this new evidence will be reflected in your updated evaluation of our work.

---

> > ### Author Response · Authors · 2025-08-06
> >
> > To follow up on our earlier response, we have since posted a global comment with a concrete, qualitative example of the explanations the framework produces (**"Example Explanation from Mastermind-463"**), as requested by other reviewers.
> >
> > We thought this might be a helpful addition for your final evaluation as well.
> >
> > We appreciate the workload of reviewing and thank you again for your time.

---

### Official Review · Reviewer_1dbC · 2025-06-27

**Clarity:** 2
**Significance:** 2
**Originality:** 2
**Rating:** 4
**Confidence:** 3

**Summary:**

This paper introduces a scalable method for approximating Shapley values for RL policies. This method can be used to quantify the influence of individual state features on the agent's behavior, its performance, and its value estimates. The paper also addresses two key challenges: computing Shapley values from off-policy data, and doing so while the policy is being trained.

**Questions:**

1. *Section 3.2:* How is the single-conditioned policy "defined"? I understand it as a policy that takes the initial action based solely on the features in $\mathcal{C}$ and then follows the actual policy afterward. But how are the probabilities computed for that first action? Are they approximated using a model trained with the loss defined in Equation 12?

2. *Section 3.2:* How is $\mathcal{B}$ obtained? The text states, "we uniformly sample $s_e$ and $\mathcal{C}$ at each decision stage." What does this mean? Does it imply the environment can be reset to any arbitrary state?

3. *Section 3.2:* Could you elaborate on how the off-policy (Equation 14) and on-policy (Equation 15) versions work? This is confusing, as in both cases $s'$ and $a$ seem to be obtained while following the current policy. Additionally, Equation 15 is described as SARSA-style (an on-policy method), yet is presented as off-policy, which is contradictory.

4. *Section 3.2* presents both an on-policy and off-policy method. Then, in Section 4, the challenge of learning the Shapley approximation from off-policy data is raised again, this time proposing importance sampling to reweight the loss and address distribution shift. Why is the first off-policy approach insufficient in this context? Is this instead referring to **offline** learning rather than off-policy learning?

5. Could you provide intuition for why the method proposed in Section 5 converges faster than the one that uses the exact behavior characteristic function?

**Ethical Concerns:**

["NO or VERY MINOR ethics concerns only"]

**Final Justification:**

The authors addressed many of my concerns and provided an example that partly demonstrates the usefulness of the proposed approach.

**Limitations:**

Although there is no specific section that explicitly discusses the limitations, some limitations are mentioned throughout the paper.

**Quality:**

2

**Strengths And Weaknesses:**

**Strengths:**

1. The paper tackles the important problem of interpreting RL policies. It starts from a computationally expensive method and proposes an approximation that improves scalability.

2. The experimental setting is sound, and the results are promising. The accuracy of the approach is compared against the exact method, showing low MSE. Moreover, the method is adapted for use with offline data collected from a different policy, and for explaining continually evolving policies, both of which are important use cases.

**Weaknesses:**

1. The introduction and background sections argue that RL poses additional challenges compared to supervised learning, particularly due to the temporal dependencies between state features and returns. However, this concern primarily applies when measuring the influence of state features on performance, not when measuring their influence on behavior or value estimation. The proposed approach for the latter two cases appears equivalent to existing supervised learning methods described in the background section. This raises the question of why these are presented as novel contributions if existing techniques can be directly applied.

2. Section 3.2 is somewhat difficult to follow, particularly the discussion of the performance characteristic function. See Questions 1–3 below.

3. Section 5 introduces a new method that, according to Figure 3c, appears more accurate and more scalable than the previously proposed method. It is unclear why this method is only introduced toward the end of the paper and not analyzed in greater depth earlier. This is especially puzzling given that, in line 180, the Monte Carlo estimation approach is dismissed for lacking scalability and generalization across different feature subsets.

4. Aside from demonstrating accuracy on a toy problem where Shapley values can be computed exactly, the paper lacks experiments that showcase the method’s usefulness for explaining the behavior of RL policies in high-dimensional and complex environments.

---

> ### Author Rebuttal · Authors · 2025-07-29
>
> Thank you for your detailed review and positive feedback. We address the points you raised below, starting with clarifying sections 3.2 and 4, which could have been clearer.
>
> **On the Conditioned Policy and Data Collection (Q1-2)**
>
> We will answer questions 1 and 2 together, as they relate to the same mechanism. To clarify, the framework *does not* assume the ability to reset the environment. The buffer $\mathcal{B}$ for the on-policy performance characteristic is collected via a continuous, online interaction.
>
> The process for collecting a single transition at decision stage $t$ is as follows:
> 1.  The agent is in its current environment state, $s_t$.
> 2.  Separately, a state-to-be-explained, $s_e$, and a feature subset, $\mathcal{C}$, are sampled (e.g. from a replay buffer and a uniform distribution, respectively). These are used as conditioning parameters for the value function we are learning, $V(s | s_e, \mathcal{C})$.
> 3.  Conditioning on the sampled $s_e$ and $\mathcal{C}$ collapses the single conditioned policy $\hat{\pi}(a|s;s_e, \mathcal{C})$ into a standard policy for this step. The agent takes an action, $a_t$, based on this policy: if its current state $s_t$ happens to be the state being explained, $s_e$, it acts according to the behaviour characteristic model (Equation 12); otherwise, it acts according to its original policy $\pi$.
> 4.  This action produces a single transition $(s_t, a_t, r_{t+1}, s_{t+1})$ in the environment, which is stored in buffer $\mathcal{B}$.
>
> This process is repeated at every decision stage, allowing us to learn a single value function $V(s | s_e, \mathcal{C})$ that is capable of estimating the expected return from *any* state $s$, given that the policy is modified at state $s_e$, for *any* state-coalition pair $(s_e, \mathcal{C})$. The values $V(s | s_e, \mathcal{C})$ for states $s$ not equal to $s_e$, for each pair $(s_e, \mathcal{C})$, are learnt to use for bootstrapping during training. However, our ultimate goal is to approximate the performance characteristic for $s_e$ and $\mathcal{C}$. This is recovered by querying the learned function with $s = s_e$, that is, $V(s_e | s_e, \mathcal{C})$. This mechanism allows us to learn the required values completely on-policy, without any environment resets.
>
> **On the On-Policy vs. Off-Policy Methods in Sec 3.2 (Q3)**
>
> We apologise for the confusing terminology, particularly "SARSA-style", which we will remove. To clarify the two methods:
>
> * The **on-policy** method (Eq. 14) is so-named because it populates its buffer $\mathcal{B}$ by collecting new experience. As just described, the entire tuple $(s, a, r, s')$ is generated *by the conditioned policy itself*.
>
> * The **off-policy** method (Eq. 15) is so-named because it uses a buffer $\mathcal{B}$ where the experience tuples $(s, a, r, s')$ were generated *by a different, arbitrary policy*. The update rule is indeed off-policy because the data source is different from the policy being evaluated. It is more similar to the off-policy algorithm Q-learning: instead of bootstrapping from the value of the *maximising* action, it bootstraps from the value of the target policy's chosen action $a'$; the similarity of using the action $a'$ was our reason for the poor choice of the "SARSA-style" terminology.
>
> **On the Off-Policy Methods in Sec 3.2 vs. Sec 4 (Q4)**
>
> This question highlights a distinction that we should have made clearer. The method in Section 3.2 is not insufficient; rather, the two sections present distinct tools for approximating different parts of the explanation framework in the off-policy setting.
>
> * The off-policy method in Section 3.2 is a specific tool designed to approximate the *performance characteristic*.
> * The importance sampling method in Section 4 is a more general technique used to approximate the *behaviour and value estimation characteristics* from a replay buffer.
>
> The separation was narrative-driven, reflecting two different scenarios for data collection. Section 3.2 considers a setting where deliberate online interaction for explanations is possible, making off-policy learning a tool for more sample-efficient data reuse. In contrast, Section 4 addresses the more constrained setting where explanations must be learnt passively from the agent's existing replay buffer.
>
> You are correct that the scenario in Section 4 is similar to "offline learning", but we prefer the more general term "off-policy learning" because "offline" typically implies a static, fixed dataset, which would be incompatible with our continual learning setting where the replay buffer is constantly updated.
>
> **On the Narrative Structure and Section 5 (Weakness 3, Q5)**
>
> Our narrative choice was to first establish a general, model-based framework to provide a solid foundation upon which extensions could be built, with the method in Section 5 being one such extension. We initially dismissed purely sampling-based methods because they are computationally expensive for repeat explanations, making them ill-suited for the core challenge of explaining non-stationary policies. In contrast, the hybrid approach in Section 5 preserves the crucial ability to continually and efficiently update explanations for evolving agents by integrating single-sample estimates directly into the parametric training.
>
> Your final question (Q5) raises an excellent point about why the sampling-based method converges faster than using the exact characteristic function. We believe this counter-intuitive result is due to the noise from the single-sample approximations acting as a *stochastic regulariser*. The noise prevents each model update from overfitting to a batch's exact targets and encourages the optimiser to learn the smoother, underlying expected function more efficiently. This phenomenon has also been explored in contemporary work on stochastic amortisation [6].
>
> **On the Originality of Behaviour and Value Estimation Approximations (Weakness 1)**
>
> We agree that, when viewed in isolation, the base methods for approximating the behaviour and value estimation characteristics are structurally similar to existing supervised learning techniques and are not, by themselves, a notable contribution.
>
> Our primary contribution is not the individual equations but the complete, integrated framework designed specifically for the practical challenges of reinforcement learning. The behaviour and value estimation approximators are necessary components to complete this framework, which is collectively an important contribution because it also addresses: (1) approximating performance explanations in a tractable way, and (2) generating explanations for non-stationary policies and from off-policy data.
>
> Furthermore, the sampling-based update (Section 5) is a significant algorithmic contribution that is also directly applicable to improving Shapley value approximations in supervised learning, demonstrating originality beyond the specific context of reinforcement learning.
>
> **On Limited Evaluation in Complex Environments (Weakness 4)**
>
> Our deliberate use of smaller domains was a direct consequence of the need to compute exact Shapley values for validation. Approximation accuracy can be rigorously tested *only* in these verifiable settings. Therefore, our experiments were necessarily constrained to this regime to ensure scientific validity.
>
> That said, while we cannot directly assess accuracy in larger domains, we agree that general scalability in more complex environments is important. Motivated by your feedback and that of others, we have run new experiments on substantially larger versions of Mastermind, designed to directly address the need for evaluation in more complex settings.
>
> While it is impossible to measure accuracy against an exact ground truth in these domains, our theoretical guarantees and the strong accuracy demonstrated in verifiable settings suggest that FastSVERL's models will converge towards the correct values. Therefore, our goal with these new experiments is to empirically test three key properties at scale: (1) if the models indeed converge, (2) the stability of that convergence across runs (measured by standard error), and (3) if the computational cost of convergence remains manageable.
>
> Due to space constraints, we present a representative subset of results below for the behaviour characteristic and Shapley models over 10 runs.
>
> **Table 1: Convergence and Stability of Behaviour Explanations**
> | Domain | Explanation Model | Updates to Converge (Mean ± Std. Err.) | Final Loss (Mean ± Std. Err.) |
> | :--- | :--- | :---: | :---: |
> | **Mastermind-443** | Behaviour Characteristic | $(1.13 \pm 0.177) \times 10^6$ | $0.0038 \pm 0.000018$ |
> | (24 features, $\ge 4.3 \times 10^7$ states) | Behaviour Shapley | $(7.99 \pm 0.902) \times 10^5$ | $0.0013 \pm 0.000037$ |
> | **Mastermind-453** | Behaviour Characteristic | $(1.30 \pm 0.150) \times 10^6$ | $0.0036 \pm 0.000021$ |
> | (30 features, $\ge 3.5 \times 10^9$ states) | Behaviour Shapley | $(7.91 \pm 0.503) \times 10^5$ | $0.0010 \pm 0.000032$ |
> | **Mastermind-463** | Behaviour Characteristic | $(1.25 \pm 0.174) \times 10^6$ | $0.0037 \pm 0.000013$ |
> | (36 features, $\ge 2.8 \times 10^{11}$ states) | Behaviour Shapley | $(6.56 \pm 0.816) \times 10^5$ | $0.0019 \pm 0.000044$ |
>
> The results in Table 1 confirm that the models successfully converge in these large-scale domains. The standard error across runs demonstrates this convergence is stable and highly reliable. Most importantly, the number of training updates required for convergence remains similar even as the domain complexity grows.
>
> We thank you again for your time and constructive feedback. We hope that our response and the new large-scale experiments will help to address your initial concerns. We will incorporate the full set of these new results, along with all the clarifications discussed, into the final version of the manuscript, and hope this new evidence will be reflected in your updated evaluation of our work.

---

> > ### Comment · Reviewer_1dbC · 2025-08-04
> >
> > I would like to thank the authors for addressing most of my concerns.
> >
> > To clarify, when I asked about experiments in larger domains, I did not mean that I was looking for results demonstrating the scalability of the method. Rather, I was interested in its usefulness. Specifically, are the behavioral insights this method provides about a particular RL agent helpful for interpretability?

---

> ### Author Response · Authors · 2025-08-06
>
> We are happy that most of your concerns have been addressed and apologise for the initial confusion regarding scalability versus usefulness. While we agree that demonstrating the qualitative value of the generated explanations is a key step, the primary focus of this paper was on the significant computational challenges of approximating Shapley values. A formal user study to rigorously evaluate their practical usefulness is an important direction for future work, but was beyond the scope of this manuscript.
>
> However, we have provided a concrete example from our new large-scale experiments in our global comment, **"Example Explanation from Mastermind-463"**, to demonstrate the kinds of interpretable, behavioural insights the framework can produce.
>
> We believe this example supports that the framework can produce plausible and interpretable insights. We thank you again for pushing us to include this qualitative example, as we agree it significantly strengthens the paper. We would be happy to include more such examples in our final revision, while emphasising that a formal user study remains an important direction for future work.

---

> > ### Comment · Reviewer_1dbC · 2025-08-06
> >
> > I want to thank the authors for providing a new example that showcases the usefulness of the approach in interpreting the behaviour of RL agents.
> >
> > I understand that the focus of the paper is on enabling the computation of Shapley values for large-scale RL. However, without evidence of its practical utility, it is difficult to argue that such an approach is truly necessary. The example provided is a first step in that direction. As such, I will raise my rating to a 4, as I believe that a few more examples would be needed to fully demonstrate its usefulness. In my view, this should have been included in the original submission.

---

> > > ### Author Response · Authors · 2025-08-06
> > >
> > > Thank you for your thoughtful follow-up and for raising your score. We are glad you found the new example helpful in showcasing the method's usefulness. We also appreciate your perspective on the broader, important discussion about the ultimate utility of different interpretability methods. We strongly believe these questions are best answered by studies conducted in real-world systems with the intended users, and we will encourage this in our paper's concluding thoughts.
> > >
> > > Thank you again for the constructive and insightful review process, which has helped to strengthen the manuscript.

---

### Official Review · Reviewer_ayeY · 2025-06-30

**Clarity:** 2
**Significance:** 3
**Originality:** 3
**Rating:** 5
**Confidence:** 4

**Summary:**

The submission studies Shapley values for explaining reinforcement learning models. Similar to FastSHAP, the submission proposes FastSVERL to amortize estimation of Shapley values for reinforcement learning as introduced in SVERL. The submission comprehensively discusses amortization for both behavior, performance, and value estimation. Experiments on both synthetic and real-world datasets show the behavior of the proposed methods and discuss its limitations and potential future research directions.

**Questions:**

* Several times throughout the paper it is stated that "Shapley value explanations do not generalize" hence are "inefficient for repeated explanations". Could the authors clarify what they mean? What does it mean for an explanation to "generalize"?

* Lines 202-204: I have a general question about explaining the performance of a model. Why is the explained state $s_e$ always the start state? How could one explain the performance of the model at any state $s$ of a trajectory which may not be the first one? Does the modified policy take into account that the trajectory may pass through state $s_e$ again later? What would happen at that point?

* Could the authors clarify what they mean by "earlier policies"? Does this assume access to the model during training to collect observations? This is an important consideration for interpretability because usually we assume being given a fixed predictive model after training is completed. If FastSVERL requires access to trajectories during training, it should be made explicit and its implications discussed.

* Could the authors expand on how ground-truth for MSE evaluation is obtained? Does "exact values" refer to exact Shapley values? Wouldn't these require expectation over an unknown distribution to mask the inputs?

* Lines 325-326: I am missing how it can be that the models start with low error if the policy is random. If the policy is random, then there is "nothing to explain", so what are the models learning?

* Does the sampling strategy apply to performance and value estimation as well?

* It might be interesting to expand on the sampling strategy, since it seems to outperform the models. After reading the last paragraph in Sec. 5, it was unclear to me whether the message of the submission is that sampling is better than training models, which can be expensive and difficult to do in practice.

---

**Minor comments**

* Abstract: "real-time" estimation feels outside the scope of the submission. Experiments do not discuss latency or role of real-time estimation for downstream tasks.
* The difference between behavior, performance, and value could be spelled out more clearly for readers coming from the interpretability community who may not be familiar with reinforcement learning.
* Lines 40-43 are too technical and hard to understand at this point in the paper.
* Line 78: throughout the paper, the cardinality of the state space $\lvert \mathcal{X} \rvert$ is mentioned several times. Is the state space assumed to be finite in this work? In several applications, this may not be finite. Same question about $\mathcal{S}$
* Line 82: it is unclear what "state-of-the-art" means in this context.
* Line 86: can $p(C)$ be any distribution or should it be a uniform to retrieve the conditional expectation?
* Eq. (8) typo in conditioning, should be $S_t = s^C$? Also $G_t$ is undefined
* Line 213: reference for "DQN-style loss"?
* Eq. (14): $V$ and $V'$ are also undefined
* Line 337: it is unclear what "Shapley models" refer to

**Ethical Concerns:**

["NO or VERY MINOR ethics concerns only"]

**Final Justification:**

The submission builds on existing methods: SVERL + FastSHAP, to provide explanations for deep reinforcement learning models, both at the performance, behavior, and value estimation levels.

The method and findings are interesting to the community, and evidence supports the claim made in the paper, hence I recommend acceptance.

It will be important to include in the revised version of the paper the main points of discussion that were raised by reviewers. In particular, larger-scale experiments, and to clarify the message between models and sampling strategies.

**Limitations:**

yes

**Quality:**

3

**Strengths And Weaknesses:**

**Strengths**

1. Explainability for reinforcement learning is a challenging setting that is worth exploring
2. Amortization has proven to be a successful strategy in the past
3. Experimental analysis is thorough

**Weaknesses**

1. Presentation could do a better job at separating behavior, performance, and evaluation estimation, which is hard to follow
2. Certain claims are hand-wavy and could be clarified
3. Sampling strategies could be given more space

I have a few clarifying questions and I am looking forward to discussing with the authors!

---

> ### Author Rebuttal · Authors · 2025-07-29
>
> Thank you for your thorough and constructive review, along with your positive assessment of our work. Your questions and comments have improved the paper's clarity significantly; we address each below.
>
> **On Presentation, Narrative, and the Sampling Strategy (Weaknesses 1 and 3, Q6-7)**
>
> We apologise that the separation between the explanation types could be clearer, and intend to improve this in the final version.
>
> We also agree that it would have been interesting to expand on the sampling strategies in Section 5 if we had more space. It is an interesting direction for future work. For this paper, however, our narrative choice was to focus more on establishing a comprehensive, model-based framework, which provides a solid foundation that the community can build upon. The sampling-based approach is therefore presented as one such promising extension.
>
> The message of our submission is not that the sampling-based method is a definitive replacement for the model-based one. While it shows efficiency and accuracy gains in our experiments, such a claim would require the same rigorous evaluation that the foundational model-based framework received across all the different settings considered in the paper.
>
> To answer your final question on this topic, a full formulation for applying the sampling-based approach to *value estimation* is detailed in Appendix F. For *performance*, while theoretically possible, generating a Monte Carlo return for each sample would likely be less sample-efficient than learning the characteristic function with the bootstrapping method in the paper—a standard motivation for algorithms such as Q-Learning.
>
> **On Your Main Questions (Q1-5)**
>
> Below, we clarify several key aspects of the framework by addressing your first five questions in the order they appear.
>
> **On the "Generalisation" of Explanations (Q1).**
> To clarify our use of the term "generalise": when we state in the paper that sampling-based estimates "do not generalise across states", we mean that a Monte Carlo approximation of a Shapley value is specific to a single input state. To approximate the Shapley value for a different state, the entire computationally expensive estimation must be repeated from scratch.
>
> In contrast, FastSVERL's parametric models, implemented as neural networks, learn a single function that shares its parameters across all inputs. This means that when approximating the Shapley values for a new state, the models can reuse what they have already learned from other similar states, rather than starting the entire estimation process over again. It is this ability to leverage shared patterns across the state and feature space that we refer to as generalisation.
>
> **On Explaining Performance at Any State (Q2).**
> We apologise for the confusing use of "start state" in the paper, by which we meant the state from which the performance characteristic's expected return is measured; that was a large oversight. We should have used the term "state-to-be-explained", $s_e$, which can be any state in a trajectory, not just the first. The proposed method is specifically designed to generate performance explanations for any state.
>
> To clarify the mechanism, the framework uses a single *conditioned policy*, $\hat{\pi}(a|s; s_e, \mathcal{C})$. This is a general policy that describes how the agent should behave in *any* current state $s$, given that the agent's policy has been modified at a specific state-to-be-explained $s_e$ by the feature subset $\mathcal{C}$ being unknown.
>
> Let's consider one specific pair $(s_e, \mathcal{C})$. For this pair, the general policy collapses into a standard policy with a simple rule: if the agent's current state $s$ matches the state being explained, $s_e$, it acts according to the behaviour characteristic model; otherwise, it follows its original policy $\pi$. This directly answers your question about what happens if the trajectory passes through $s_e$ again: this is accounted for, and the agent will indeed act according to the behaviour characteristic model again.
>
> From this policy, we can learn a single value function, $V(s | s_e, \mathcal{C})$, which estimates the expected return from *any* state $s$, given that the policy is modified at a particular state $s_e$ with unknown features $\mathcal{C}$. While we also learn the values for states $s$ not equal to $s_e$, for bootstrapping during training, our ultimate goal is to approximate the performance characteristic for the pair $(s_e, \mathcal{C})$. This is recovered by querying the learned function at the particular state $s = s_e$, that is, $V(s_e | s_e, \mathcal{C})$.
>
> Since this entire process can be applied to any state-coalition pair, the single learned value function allows us to estimate the performance characteristic for any state in an environment that we wish to explain.
>
> **On 'Earlier Policies' and Training Access (Q3).**
> Thank you, we are happy to clarify this critical point about the framework's flexibility. To be clear, the framework *does not* assume access to the model during training to collect observations and can be applied post-hoc to a fixed, trained model.
>
> The comment you refer to in line 209 specifically concerns the choice between the two methods provided for approximating the performance characteristic:
>
> * The **on-policy method** (Eq. 14) is designed for precisely the standard scenario you describe. Assuming one is given a fixed predictive model after training is complete, explanations are generated post-hoc by using on-policy data from the trained model.
>
> * The **off-policy method** (Eq. 15) provides an alternative option. In this case, "earlier policies" refers to the policies that generated a pre-existing buffer of data. Accessing such a buffer is common practice for reinforcement learning agents, and this method requires only the historical *data*, not the models that generated it.
>
> We deliberately provide both methods to make the framework general. It fully supports the standard post-hoc explanation setting you describe. The off-policy approach then extends this by opening the door to greater sample efficiency and, crucially, the ability to explain agents *during* the training process itself. This makes the proposed framework more flexible and less restrictive than the standard post-hoc-only paradigm.
>
> **On Obtaining Ground-Truth for MSE (Q4).**
> Yes, "exact values" refers to exact Shapley values. You are correct that computing these requires taking an expectation over the steady-state distribution to "mask" the unknown features. Fortunately, for the tractable domains used in our validation experiments, this conditional distribution can be accurately estimated from experience. The required expectation for the characteristic values can then be computed directly. These exact characteristic values are then plugged into the Shapley value formula to yield the final ground-truth attributions. This is the same approach used to compute exact values in the original work on SVERL [1, 2].
>
> **On Low Error for a Random Policy (Q5).**
> For an initial random (e.g. uniform) policy, the agent's actions and value estimates are independent of the state features. Therefore, the true Shapley value for any feature's contribution is zero. A randomly initialised neural network with small weights naturally outputs values close to zero, and can very quickly learn to predict zero for all inputs. The model thus starts with a low error because its initial predictions are already very close to the ground-truth attributions of zero.
>
> **On Your Minor Comments**
>
> Thank you for these comments, which will help to improve the paper's clarity. We will address each in the order they appear.
>
> * **Abstract ("real-time"):** By "real-time", we mean that explanations can be generated continually as an agent learns and acts, at a cost equivalent to selecting actions: a single forward pass through a network. We will clarify this and reconsider our wording in the final version.
> * **Difference between explanations:** This is a great point. We agree and will expand this section in the final version to be clearer for our readers.
> * **Lines 40-43:** Thank you, that's a good point, and we are happy to revise.
> * **Line 78 (Finite state space):** To clarify, the framework does not assume a finite state space and is generally applicable to either setting. The primary challenge in infinite state spaces is likely to be approximating the steady-state distribution. We discuss this in the supplement, but will clarify it in the main body of the paper.
> * **Line 82 ("State-of-the-art"):** We agree this is vague and will change the phrasing to "one approach" in the final version.
> * **Line 86 (Distribution $p(\mathcal{C})$):** Theoretically, any distribution $p(\mathcal{C})$ with full support over the feature subsets can be used in Equation 4. This is clarified in the appendix proofs for the characteristic functions we proposed.
> * **Eq. (8) typo:** Thank you for the close reading. The conditioning on $s$ is correct, as $\tilde{v}_{s}^{\pi}(\mathcal{C})$ is the expected return measured from that specific state $s$. We apologise that the return $G_t$ was undefined and will fix this oversight.
> * **Line 213 (DQN-style loss):** Thank you, we will add the reference.
> * **Eq. (14) undefined terms:** The value function $V$ is defined in line 206, and the target network $V'$ is introduced in line 216. However, the specific role of the target network could be more explicit; we will clarify this in our revision.
> * **Line 337 ("Shapley models"):** This is our shorthand for the parametric models trained to predict Shapley values. We will define this term explicitly to avoid confusion.
>
> Thank you again for your constructive review, which helped improve the paper's clarity in several areas. We hope our response helps to address your initial concerns. We will incorporate these clarifications into the final manuscript, and we hope this will be reflected in your updated evaluation of our work.

---

> > ### Comment · Reviewer_ayeY · 2025-08-03
> > **Thank you for your response!**
> >
> > I sincerely thank the authors for their detailed responses and thoughtful consideration of all reviewers' comments.
> >
> > The authors have answered all my concerns, and I ask that the main points on clarity of presentation be included in the revised version of the paper to make sure the overall narrative is communicated effectively.
> >
> > ---
> >
> > **Experiments on realistic datasets**
> >
> > I commend the authors for their consideration of several reviewers' concerns regarding applicability of the proposed method to more realistic scenarios. I agree that exploring experiments with 24+ features is compelling within the context of Shapley value estimation.
> >
> > However, I am wondering whether it would be feasible for the authors to provide some example explanations produced by their method in these experiments? I think this would be an important complement to the table included in the rebuttal that would strengthen evidence in these scenarios where exact estimation is not possible, and convergence cannot be interpreted as a notion of correctness on its own.

---

> ### Author Response · Authors · 2025-08-06
>
> Thank you for your follow-up. We are pleased that our response has addressed your concerns.
>
> You have raised a great point, and we agree that a qualitative example is a nice complement to the new large-scale quantitative convergence results. We have provided a concrete demonstration in our global comment, titled **"Example Explanation from Mastermind-463"**, which we hope provides the kind of evidence you are looking for.
>
> We will, of course, incorporate the main points on clarity, the new large-scale results, and additional qualitative examples into the revised version of the paper. Thank you again for your valuable feedback, which has significantly strengthened the manuscript.

---

> > ### Comment · Reviewer_ayeY · 2025-08-06
> > **Thank you for your reponse!**
> >
> > I sincerely thank the authors for including an example illustration of the kind of explanations produced by their methods. These will be important to include in the revised version of the manuscript.
> >
> > I raised my score to reflect discussion with the authors.

---

### Official Review · Reviewer_boW8 · 2025-07-02

**Clarity:** 3
**Significance:** 4
**Originality:** 3
**Rating:** 5
**Confidence:** 4

**Summary:**

Shapley values are widely used to provide interpretability to RL behaviors, however, their exponential computation cost makes them difficult to compute in many real problems.
The manuscript introduces FastSVERL, a scallable method for approximating Shapley values in the context of RL. It takes ideas from FastSHAP and extends them to the RL context. It particularly deals with sequential, policy-based RL environments, handling temporal dependencies and off-policy learning.

**Questions:**

- How does the method perform in large-scale, high-dimensional RL benchmarks beyond the toy domains used?
- What is the overhead introduced by calculating the parametric model? What is its O shape?
- Can FastSVERL be applied to any RL problem? Are there any requirements in terms of discrete/continuous domains? Continuous or episodic? Fully observable or partially observable?...
- Can parametric models be computed in all cases? What amount of data is needed?

**Ethical Concerns:**

["NO or VERY MINOR ethics concerns only"]

**Final Justification:**

Authors have successfully addressed the raised concerns and appropriately engage in insightful discussions, and thus, I reiterate my original ranking of acceptance.

**Limitations:**

At the end of the paper, the authors state the main limitation of this work: its practical deployment in real-world to test its robustness is missing.
There is no potential negative societal impact of this work.

**Paper Formatting Concerns:**

no concerns

**Quality:**

3

**Strengths And Weaknesses:**

STRENGTHS:
- The proposal is very well motivated (need for practical scalable ways of providing interpretability in RL).
- Specifically deals with RL challenges such as temporal dependencies, evolving behaviours, and learning from off-policy data.
- The performed experimentation (in the supplementary material) is very solid, analyzing accuracy, efficiency and scalability.

WEAKNESSES:
- While the goal is to provide explainability to complex RL scenarios, the given experiments are only on moderate-complexity tractable problems.
- As reported by the authors, the manuscript establishes a foundation for scalable Shapley-based explanation in RL, but the practical deployment is yet to be fully done.
- The proposal relies on parametric models, whose generalization capabilities can be limited.

---

> ### Author Rebuttal · Authors · 2025-07-29
>
> Thank you for your supportive and positive review. We are grateful for your kind words on the paper's motivation, the handling of reinforcement learning challenges, and the solid experimentation. We are happy to address the points and clarify the questions you raised.
>
> **On Scalability and Practical Deployment (Weakness 1, 2 & Q1)**
>
> We agree that demonstrating how the proposed methods scale to more complex, large-scale benchmarks is an important next step. Our use of smaller domains was a direct consequence of the need to compute exact Shapley values for validation. This is a crucial methodological point: any claims about approximation accuracy can be rigorously tested only in these verifiable settings. Therefore, our experiments were necessarily constrained to this regime to ensure scientific validity.
>
> While we cannot directly assess accuracy in larger domains, we were motivated by your feedback and that of others to provide further evidence of the framework's scalability. We have run new experiments on substantially larger versions of Mastermind. We believe these new domains, the largest having 36 features and over 280 billion states, move beyond the "toy" setting, particularly from a Shapley approximation perspective where the summation is over $2^{36}$ feature subsets.
>
> The theoretical guarantees and strong accuracy demonstrated in verifiable settings suggest that FastSVERL's models will converge towards to the correct values. Therefore, our goal with these new experiments is to test three key properties at scale: (1) if the models indeed converge, (2) the stability of that convergence across runs (measured by standard error), and (3) if the computational cost of convergence remains manageable.
>
> Due to space constraints, we present a representative subset of results below for the behaviour characteristic and Shapley models, reporting the mean training updates to convergence and the final loss on their training objectives over 10 runs. It is important to clarify that this loss is not a measure of ground-truth accuracy. The characteristic model's loss is designed to converge to a non-zero value as it learns to predict an expectation over the unknown features. In contrast, the Shapley model's loss is designed to converge to zero, but this indicates it has successfully learned to explain the values provided by the characteristic model, not necessarily the ground truth.
>
> **Table 1: Convergence and Stability of Behaviour Explanations**
> | Domain | Explanation Model | Updates to Converge (Mean ± Std. Err.) | Final Loss (Mean ± Std. Err.) |
> | :--- | :--- | :---: | :---: |
> | **Mastermind-443** | Behaviour Characteristic | $(1.13 \pm 0.177) \times 10^6$ | $0.0038 \pm 0.000018$ |
> | (24 features, $\ge 4.3 \times 10^7$ states) | Behaviour Shapley | $(7.99 \pm 0.902) \times 10^5$ | $0.0013 \pm 0.000037$ |
> | **Mastermind-453** | Behaviour Characteristic | $(1.30 \pm 0.150) \times 10^6$ | $0.0036 \pm 0.000021$ |
> | (30 features, $\ge 3.5 \times 10^9$ states) | Behaviour Shapley | $(7.91 \pm 0.503) \times 10^5$ | $0.0010 \pm 0.000032$ |
> | **Mastermind-463** | Behaviour Characteristic | $(1.25 \pm 0.174) \times 10^6$ | $0.0037 \pm 0.000013$ |
> | (36 features, $\ge 2.8 \times 10^{11}$ states) | Behaviour Shapley | $(6.56 \pm 0.816) \times 10^5$ | $0.0019 \pm 0.000044$ |
>
> The results in Table 1 confirm that the models successfully converge in these large-scale domains. The standard error across multiple runs demonstrates that this convergence is stable and highly reliable. Most importantly, the number of training updates required for convergence remains similar even as the domain complexity grows.
>
> We also demonstrate that the significant computational advantages of the sampling-based approximations we propose (Section 5) hold at scale. This method is a key contribution of our work, and Table 2 confirms its efficiency benefits in these large domains.
>
> **Table 2: Total Computational Cost for Behaviour Explanations**
> | Domain | Total Updates (Model-Based) (Mean ± Std. Err.) | Total Updates (Sampling-Based) (Mean ± Std. Err.) |
> | :--- | :---: | :---: |
> | **Mastermind-443** | $(1.84 \pm 0.136) \times 10^6$ | $(5.69 \pm 0.498) \times 10^5$ |
> | **Mastermind-453** | $(2.09 \pm 0.095) \times 10^6$ | $(7.25 \pm 0.638) \times 10^5$ |
> | **Mastermind-463** | $(1.91 \pm 0.104) \times 10^6$ | $(6.06 \pm 0.584) \times 10^5$ |
>
> By eliminating the need to pre-train a characteristic model, the sampling-based approach reduces the total computational cost by over $2\times$ across all large-scale domains tested. This suggests that the substantial efficiency gains are consistent.
>
> While we agree that the practical deployment of FastSVERL is a key step towards validating its effectiveness, this is beyond the scope of the current manuscript, and we hope instead these new results will help to address your questions about scalability.
>
> **On Computational Overhead and Complexity (Q2)**
>
> Thank you for this question. The main computational overhead of the framework is the training of the parametric models. Once trained, the cost to generate an explanation for any given state is a single, efficient forward pass through the networks.
>
> For the standard case of explaining a fixed, stationary policy post-hoc, the training overhead consists of two main parts:
> 1.  **Data Collection:** This is the cost of collecting experience to approximate the steady-state distribution. This can be running the fixed policy (on-policy) or mitigated by reusing a pre-existing data buffer (off-policy).
> 2.  **Model Training:** This is the cost of training the parametric models. For the behaviour and value estimation characteristics and the Shapley models, this is analogous to a standard supervised learning training cost. For the performance characteristic, it is that of a standard reinforcement learning process.
>
> In the setting where the explanation models are updated continuously alongside the agent's policy, the only additional computational cost beyond the agent's standard policy updates is the cost of updating the explanation models, which for each model is roughly equivalent to the agent's own update cost at each training step.
>
> Regarding the complexity (`O` shape), providing a single theoretical `O` notation for the training of deep models is difficult, as it depends on many hyperparameters. However, the Hypercube experiments (Figure 2 in the paper and supplement) provide some empirical evidence of the framework's scaling properties. The results show that in this verifiable setting, the number of training updates required for convergence scales approximately polynomially, as opposed to the exponential complexity of computing Shapley values exactly.
>
> **On General Applicability and Requirements (Q3)**
>
> This is an excellent question, and we're glad you prompted us to clarify the framework's scope. We will add a more detailed discussion of these points to the final paper.
>
> In general, the framework applies to any problem where an agent observes, acts, and receives rewards for those actions. To be more specific:
> * **State/Action Spaces:** The work in this paper focuses on discrete action spaces because the behaviour characteristic is defined over them. However, the theoretical SVERL framework provides explanations for continuous action spaces, and the proposed approximation methods can be readily adapted. The parametric nature of the models also allows the framework to be applied to continuous state spaces; the primary challenge in such settings becomes approximating the steady-state distribution, a point we discuss below and in the supplement, which we will move to the main body.
> * **Episodic/Continuous Tasks:** The framework applies to both episodic and continuous tasks.
> * **Observability:** The framework can be applied in partially observable settings, where attributions would be over the features of the agent's observation. This setting is particularly interesting for explaining non-stationary policies, as such policies are often a direct consequence of partial observability.
>
> **On Generalisation and Data Requirements (Weakness 3 & Q4)**
>
> Regarding the limitations of parametric models, you raise a valid point about the general challenges of out-of-distribution generalisation. However, our primary goal in this work is amortisation, which is a form of in-distribution generalisation. Neural networks are well-suited for this task, as they learn to interpolate within the training distribution, allowing them to generalise across the vast combinatorial space of states and feature subsets. Furthermore, a parametric approach was a necessary choice to enable the continual learning aspect of the framework, as non-parametric sampling methods cannot be efficiently updated to handle non-stationary policies.
>
> Your question about data requirements is also very important. The practical applicability of the framework depends on collecting enough data to form a reasonable approximation of the relevant steady-state distributions. The amount of data required is analogous to that needed to learn a value or policy function in the same domain, as all three tasks fundamentally rely on the experience providing a sufficient sample of the underlying state visitation distribution. This is a key challenge for future work, especially in large or continuous state spaces where a frequentist approximation with a buffer is necessarily a sparse sample of the true distribution. We briefly discuss this important topic in the supplement, but will move it to the main body in our revision.
>
> Thank you again for your supportive review and helpful feedback, and questions; they have helped to clarify many important aspects of the framework, and we will incorporate all of these clarifications and our new results into the final version of the paper. We hope this reinforces your positive assessment of our work.

---

> ### Comment · Reviewer_boW8 · 2025-08-06
>
> Thanks for the responses to my questions and those of my fellow reviewers. I think that they are appropriately addressed and I reiterate my ranking of acceptance. I encourage the authors to perform the additional experiments that have been requested so that the paper can be improved.
>
> Also, please do not forget to include the modifications indicated in the updated version of the paper, should it be finally accepted.

---

> > ### Author Response · Authors · 2025-08-06
> >
> > Thank you very much for your time and for confirming your support for our paper. We will certainly include the additional experiments and all clarifications in the final manuscript. We are very grateful for your helpful feedback throughout this process.

---

### Official Review · Reviewer_2mtF · 2025-07-03

**Clarity:** 3
**Significance:** 2
**Originality:** 2
**Rating:** 4
**Confidence:** 3

**Summary:**

This paper introduces a framework for computing approximate shapley values to explain reinforcement learning policies. In particular, this work uses parametric models to amortize the approximation cost of shapley values across states and features which is compatible with both on-policy and off-policy methods. Explanations are generated to explain an agenet's behavior (policy), performance (reward), and value (future reward).

**Questions:**

1. Are the estimators used in this paper unbiased?

2. There is little analysis on the single-sample approximation of characteristic values. I would think that this would increase variance given that these are Monte Carlo samples. Is this the case? If not, why?

**Ethical Concerns:**

["NO or VERY MINOR ethics concerns only"]

**Final Justification:**

The authors have clarified many of my concerns about the paper. Although I'm still not completely sold on Mastermind as the core evaluation benchmark, I think their claims of a computationally efficient method to calculate Shapley values are fairly well supported so I am updating my score from 3 to 4.

**Limitations:**

Yes.

**Quality:**

3

**Strengths And Weaknesses:**

Strengths:
* Fast and efficient shapley-based explanations can be useful for a wide-variety of tasks, particularly given the heavy application of RL to many real-world problems.

* The proposed method supports both off-policy and on-policy learning, the latter of which is particularly useful as it supports explanations for non-stationary policies. Explaining how policies change during the course of training could be very useful.


Weaknesses:
* Experiments are only performed over very simple environments (a 2x4 Gridworld in the Appendix, and the Mastermind code-breaking game with codes of up to length 3). While I understand that the main results in the paper used small environments so as to support exact value calculation, this means that the *scalability* of the method is not tested. This is particularly relevant given that the paper claims that FastSVERL is suitable for real-time and real-world problems, yet the evaluation is only conducted over small toy problems that are not at all representative of real-world ones. Does this method scale to more complex environments/policies with respect to both computation time and approximation accuracy?

* No mention is given to bias or fairness of the approximations. Are the estimators used in this paper unbiased?

* There is little analysis on the single-sample approximation of characteristic values. I would think that this would increase variance given that these are Monte Carlo samples. Is this the case? If not, why?

---

> ### Author Rebuttal · Authors · 2025-07-29
>
> Thank you for your time and constructive feedback. We are glad you appreciate the importance of fast and efficient Shapley-based explanations and see the value in developing approximations tailored to RL-specific settings.
>
> We address each of your points on scalability, estimator bias, and the variance of the single-sample approximation below. We hope our clarifications help to resolve your concerns.
>
> **On Scalability and Evaluation in "Toy" Environments (Weakness 1)**
>
> This is an important point. We agree that the real-world applicability of a method is essential, and scalability is an important part of that. We should clarify that our use of "real-world" and "real-time" primarily refers to the framework's ability to handle the practical constraints of real-world reinforcement learning. For us, this means providing explanations for a continually evolving agent in real-time, which requires adapting explanations to non-stationary policies and learning from off-policy data. These are both central features of FastSVERL.
>
> Regarding our experimental design, you rightly point out that our deliberate use of smaller domains was a direct consequence of the need to compute exact Shapley values for validation. This is a crucial methodological point: any claims about approximation accuracy can be rigorously tested *only* in these verifiable settings. Therefore, our experiments were necessarily constrained to this regime to ensure scientific validity.
>
> It is only within this verifiable regime that we can demonstrate how FastSVERL’s accuracy scales. Our Hypercube experiments were designed specifically for this purpose, and they showed that the computational cost of achieving accurate explanations scales well with both the number of states and features. This graceful scaling of accuracy, in the one region where it can be principally measured, suggests the method should continue to perform well in larger domains. Furthermore, the results on Mastermind-333 in our supplement—with over 100,000 states and 15 features ($2^{15}$ subsets)—already push the boundary of what can be considered a "toy" domain for Shapley value computation, providing further evidence of the framework's capabilities.
>
> That said, while we cannot directly assess accuracy in larger domains, we agree that general scalability deserves greater focus. Motivated by your feedback and that of others, we have run new experiments on substantially larger versions of Mastermind. We continue to use Mastermind specifically to allow fine-grained control over complexity. We believe these new domains, the largest having 36 features and over 280 billion states, move beyond the "toy" setting, particularly from a Shapley approximation perspective where the summation is over $2^{36}$ feature subsets.
>
> While it is impossible to measure accuracy against an exact ground truth in these domains, our theoretical guarantees and the strong accuracy demonstrated in verifiable settings suggest that FastSVERL's models will converge towards the correct values. Therefore, our goal with these new experiments is to empirically test three key properties at scale: (1) if the models indeed converge, (2) the stability of that convergence across runs (measured by standard error), and (3) if the computational cost of convergence remains manageable.
>
> Due to space constraints, we present a representative subset of results below for the behaviour characteristic and Shapley models, reporting the mean training updates to convergence and the final loss on their training objectives over 10 runs. It is important to clarify that this loss is for the training objective and is not a measure of ground-truth accuracy. The characteristic model's loss is designed to converge to a non-zero value as it learns to predict an expectation over the unobserved features. In contrast, the Shapley model's loss is designed to converge to zero, but this indicates it has successfully learned to explain the values provided by the characteristic model, not necessarily the ground truth.
>
> **Table 1: Convergence and Stability of Behaviour Explanations**
> | Domain | Explanation Model | Updates to Converge (Mean ± Std. Err.) | Final Loss (Mean ± Std. Err.) |
> | :--- | :--- | :---: | :---: |
> | **Mastermind-443** | Behaviour Characteristic | $(1.13 \pm 0.177) \times 10^6$ | $0.0038 \pm 0.000018$ |
> | (24 features, $\ge 4.3 \times 10^7$ states) | Behaviour Shapley | $(7.99 \pm 0.902) \times 10^5$ | $0.0013 \pm 0.000037$ |
> | **Mastermind-453** | Behaviour Characteristic | $(1.30 \pm 0.150) \times 10^6$ | $0.0036 \pm 0.000021$ |
> | (30 features, $\ge 3.5 \times 10^9$ states) | Behaviour Shapley | $(7.91 \pm 0.503) \times 10^5$ | $0.0010 \pm 0.000032$ |
> | **Mastermind-463** | Behaviour Characteristic | $(1.25 \pm 0.174) \times 10^6$ | $0.0037 \pm 0.000013$ |
> | (36 features, $\ge 2.8 \times 10^{11}$ states) | Behaviour Shapley | $(6.56 \pm 0.816) \times 10^5$ | $0.0019 \pm 0.000044$ |
>
> The results in Table 1 confirm that the models successfully converge in these large-scale domains. The standard error across multiple runs demonstrates that this convergence is stable and highly reliable. Most importantly, the number of training updates required for convergence remains similar even as the domain complexity grows.
>
> We also demonstrate that the significant computational advantages of the sampling-based approximations we propose (Section 5) hold at scale. This method is a key contribution of our work, and Table 2 confirms its efficiency benefits in these large domains.
>
> **Table 2: Total Computational Cost for Behaviour Explanations**
> | Domain | Total Updates (Model-Based) (Mean ± Std. Err.) | Total Updates (Sampling-Based) (Mean ± Std. Err.) |
> | :--- | :---: | :---: |
> | **Mastermind-443** | $(1.84 \pm 0.136) \times 10^6$ | $(5.69 \pm 0.498) \times 10^5$ |
> | **Mastermind-453** | $(2.09 \pm 0.095) \times 10^6$ | $(7.25 \pm 0.638) \times 10^5$ |
> | **Mastermind-463** | $(1.91 \pm 0.104) \times 10^6$ | $(6.06 \pm 0.584) \times 10^5$ |
>
> By eliminating the need to pre-train a characteristic model, the sampling-based approach reduces the total computational cost by over $2\times$ across all large-scale domains tested. This suggests that the substantial efficiency gains are consistent.
>
> We hope these clarifications and new results will help to address your concerns about scalability.
>
> **On the Bias of the Estimators (Weakness 2, Q1)**
>
> Thank you for this question. To clarify, the estimators for the behaviour and value estimation characteristics are asymptotically unbiased, as are all of our Shapley estimators. These properties stem directly from their learning objectives, as proven in the supplementary material.
>
> First, our characteristic models for behaviour and value estimation are trained to minimise an expected squared-error loss. The unique global minimiser of this objective is the true conditional expectation (Appendix B). An estimator that converges to this true mean is, by definition, asymptotically unbiased.
>
> Second, all of our Shapley models are trained to solve a weighted least-squares problem whose unique solution is the true Shapley value vector for the given characteristic function (Appendix A). Therefore, these estimators are also asymptotically unbiased. The same property holds for our sampling-based objective, as we prove in Appendix F and address in the following point.
>
> **On the Variance of the Single-Sample Approximation (Weakness 3, Q2)**
>
> This is a great question. Your intuition is correct: using a single Monte Carlo sample at each training step does introduce variance into the gradient updates (though one could reduce this variance by increasing the number of samples per update).
>
> However, the key insight is that our parametric Shapley model is designed to handle this, effectively averaging out the variance over the entire training process. By optimising over many of these noisy but computationally cheap samples, the model learns to predict the expected value, guided by an objective that, as we prove in Appendix F, is unbiased in expectation and identical to the original model-based loss.
>
> This approach presents a favourable trade-off: we accept higher variance in individual gradient updates to completely eliminate the propagation of approximation error from a pre-trained characteristic model. The empirical results strongly support this design choice. As shown in Figure 3c in the main paper and the results in Figure 7 of the supplement, the sampling-based approach converges significantly faster and to a lower final error than the model-based approach. This is precisely because it avoids compounding the approximation errors from the upstream characteristic model, leading to a more accurate and efficient final result.
>
> We thank you again for your detailed and constructive feedback. We hope that our response and the new large-scale experiments will help to address your initial concerns. We will incorporate the full set of these new results, along with all the clarifications discussed, into the final version of the manuscript, and hope this new evidence will be reflected in your updated evaluation of our work.

---

> > ### Comment · Reviewer_2mtF · 2025-08-06
> >
> > Thank you for the thorough response. I think the paper would benefit from the additional experiments in a larger state space and encourage their inclusion (in the appendix or otherwise), and I appreciate the clarification on my other questions.
> >
> > While I'm not entirely convinced by Mastermind as a proxy for real-world problems, nor of the usefulness of Shapley values for the interpretability of RL agents (as complex behaviors are rarely easily describable through feature attribution techniques such as Shapley values), the paper's claims are in general supported in terms of a more computationally-efficient way to calculate values. As such, I am willing to increase my score.

---

> > > ### Author Response · Authors · 2025-08-06
> > >
> > > Thank you for engaging with our rebuttal, and we are grateful that you are willing to raise your score. We appreciate you looking past your reservations to see the paper's core contribution: a computationally-efficient way to approximate these values.
> > >
> > > You have raised an important point about the ultimate usefulness of Shapley values for interpreting complex reinforcement learning agents. We agree that feature attribution is just one piece of the puzzle and that a deep understanding of agents will require a multi-faceted approach. There is still a long way to go for the field to understand which explanation methods work best and where, but rigorous user studies and embedding explanations into real-world systems are almost certainly a key part of this.
> > >
> > > While we believe Shapley values have a valuable role to play, we respect that the community is still exploring their utility. If you are interested, we have posted a qualitative example in a global comment ("Example Explanation from Mastermind-463") that demonstrates the kinds of behavioural insights one might be able to draw from them.
> > >
> > > Thank you again for your thoughtful consideration and for helping us to clarify the scope and contributions of our work.

---

### Author Response · Authors · 2025-08-06
**Example Explanation from Mastermind-463**

In response to requests for a qualitative example of the framework's usefulness, we provide a concrete example from our new large-scale experiments demonstrating the kinds of interpretable, behavioural insights the framework can produce.

To provide context, Mastermind-463 is a code-breaking game where an agent has six guesses to find a hidden code of length four, drawn from an alphabet of three letters. After each guess, the agent receives two clues: **Clue 2** for the number of correct letters in the correct position, and **Clue 1** for the number of correct letters in the wrong position. Full details of gameplay are provided in the paper.

The table below shows a board state from the trained Mastermind-463 agent after it has made four guesses. The agent's next chosen action, `(C, C, C, B)`, is shown in green for reference. The state is represented by the visible game board, which consists of 36 features corresponding to the previous guesses, their associated feedback, and the remaining empty guess slots.

The background colour of each cell represents the Shapley value for that feature, as predicted by the trained behaviour explanation model. For this visualisation, we have mapped these numerical values to a colour scale where the intensity of the blue indicates the strength of the positive influence on the agent's decision.

$\definecolor{white}{RGB}{255, 255, 255}$
$\definecolor{l_blue}{RGB}{188, 215, 243}$
$\definecolor{m_blue}{RGB}{125, 180, 234}$
$\definecolor{s_blue}{RGB}{74, 144, 226}$
$\definecolor{d_green}{RGB}{28, 140, 52}$

| | **Clue 1** | **Pos 1** | **Pos 2** | **Pos 3** | **Pos 4** | **Clue 2** |
| :---: | :---: | :---: | :---: | :---: | :---: | :---: |
| **Guess 6** | $\colorbox{white}{~}$ | $\colorbox{white}{~}$ | $\colorbox{white}{~}$ | $\colorbox{white}{~}$ | $\colorbox{white}{~}$ | $\colorbox{white}{~}$ |
| **Guess 5** | $\colorbox{white}{~}$ | $\colorbox{white}{\textcolor{d_green}{C}}$ | $\colorbox{white}{\textcolor{d_green}{C}}$ | $\colorbox{white}{\textcolor{d_green}{C}}$ | $\colorbox{white}{\textcolor{d_green}{B}}$ | $\colorbox{white}{~}$ |
| **Guess 4** | $\colorbox{l_blue}{2}$ | $\colorbox{m_blue}{B}$ | $\colorbox{s_blue}{C}$ | $\colorbox{m_blue}{C}$ | $\colorbox{l_blue}{C}$ | $\colorbox{l_blue}{2}$ |
| **Guess 3** | $\colorbox{white}{2}$ | $\colorbox{l_blue}{C}$ | $\colorbox{m_blue}{B}$ | $\colorbox{white}{C}$ | $\colorbox{l_blue}{C}$ | $\colorbox{white}{2}$ |
| **Guess 2** | $\colorbox{white}{0}$ | $\colorbox{white}{C}$ | $\colorbox{l_blue}{C}$ | $\colorbox{l_blue}{C}$ | $\colorbox{white}{C}$ | $\colorbox{l_blue}{3}$ |
| **Guess 1** | $\colorbox{white}{0}$ | $\colorbox{white}{A}$ | $\colorbox{white}{A}$ | $\colorbox{white}{C}$ | $\colorbox{white}{A}$ | $\colorbox{white}{1}$ |

**Interpretation**

Heeding the warnings from the original SVERL work [1, 2] on the dangers of over-interpretation, and given the constraints of the rebuttal format, we will not use our interpretation of the Shapley values to draw firm conclusions about the agent's decision-making process. Instead, we will observe some interesting properties from this example that appear to align with a logical, high-level strategy for the game:

* **From an interpretability perspective:** The most influential features (the blue cells) are from recent guesses (2-4), which are sufficient to deduce that the code is a permutation of `(B, C, C, C)`. Conversely, features from the oldest guess (Guess 1), which contained superseded information, have a relatively neutral influence. This suggests the agent's decision is driven by recent, salient information while ignoring older, less relevant evidence.

* **From an accuracy perspective:** The neutral influences for the features for the unused guess slot (Guess 6) prompted us to investigate further. We discovered that the optimal policy in this domain does not require this final guess, meaning these features are truly irrelevant to the agent's decisions. Not only was this a new insight, but it also suggests the approximation is correctly satisfying the nullity axiom of Shapley values (assigning zero value to unused features), which is a good indication that it is learning correctly.

We believe this example supports that the framework can produce plausible and interpretable insights.

---

### Author Response · Authors · 2025-08-08

We thank all the reviewers for their time, their detailed and thoughtful feedback, and their active engagement during the rebuttal period. This has been a great example of the value of the peer-review process, which has certainly strengthened the paper.

Based on the feedback, we commit to incorporating all the clarifications discussed in our responses. The main changes are summarised below:

**Clarifications and Presentation:**
* We will improve the overall clarity of the presentation and the discussion of our narrative choices.
* We will be more precise with our use of the terms "real-world" and "real-time" to avoid interpretations of overclaiming.
* We will add more detailed discussions on key technical aspects, including the unbiasedness of the estimators, the variance of the single-sample approximation, the framework's general applicability, and the data and computational requirements.
* We will move the important discussion on the challenges of approximating the steady-state distribution from the supplement to the main body of the paper.

**Additional Experimental Results:**
* We will include the new large-scale experiments and qualitative examples discussed in the rebuttal.

It is important to note that these new results do not change the paper's core thesis. They serve as an additional dimension of evidence for the framework's capabilities, reinforcing the original contribution with new results on scalability and interpretability. The core contribution of the paper remains the same: providing parametric approximations of Shapley values designed for the unique challenges of reinforcement learning.

Thank you once again for your time and for helping us to improve our work.

---

### Decision · Program_Chairs · 2025-09-17

**Decision:**

Accept (poster)

**Comment:**

The paper proposes a method for efficiently computing approximate Shapley values in RL. It builds on existing methods and extends these ideas to RL. The efficacy of the method is experimentally evaluated. The studied problem is interesting and important for explainable RL. While the reviewers initially expressed concerns about this submission---some suggesting that further experimentation is needed to demonstrate the utility of the approach in more complex settings---most concerns were addressed during the discussion, where the authors provided additional experimental results and clarified different aspects of their work. The authors should incorporate the new results and improve the quality and clarity of the presentation in the final version of the paper.